# A New Single-Cell Hybrid Inductor-Capacitor DC-DC Converter for Ultra-High Voltage Gain in Renewable Energy Applications

**Ammar Falah Algamluoli * and Xiaohua Wu ***

School of Automation, Northwestern Polytechnical University, Xi'an 710060, China
* Correspondence: ammar.algamluoli@coeng.uobaghdad.edu.iq (A.F.A.); wxh@nwpu.edu.cn (X.W.)

**Abstract:** In this paper, a new single-cell hybrid switched inductor DC-DC converter is proposed to demonstrate the verification of ultra-high voltage gain in renewable energy applications (REA). The modification involves adding a single cell of an inductor with a diode and double capacitor to increase voltage transfer gain. Additionally, this modification helps prevent the input current from becoming zero, pulsating at very low duty cycles. The single cell of the hybrid inductor is interleaved with the main switch to reduce current stress when the capacitor of the single-cell inductor charge becomes zero. Moreover, the addition of a modified hybrid switch inductor with a capacitor, operating in dual boosting mode with a single switch, allows the converter to achieve ultra-high voltage gain. The proposed converter offers several advantages, including ultra-high voltage gain, high efficiency, low voltage stress on power MOSFETs, diodes, inductors, and capacitors, as well as low switching and conduction losses. Furthermore, the proposed converter utilizes transformerless and non-coupled inductors. Mathematical equations have been derived for the discontinuous conduction mode (DCM) and continuous conduction mode (CCM) and implemented using Matlab Simulink software to validate the results. In addition, a dual PI controller is designed for the proposed converter to verify the fixed output voltage. Experimental results have also been obtained for a 200 W prototype, with the input voltage varying between 20 V and 40 V, and an output voltage of 200 V at an efficiency of 96.5%.

**Keywords:** DC-DC converter; hybrid switch inductor; single cell switched inductor; ZCS





## 1. Introduction

The world is increasingly interested in utilizing various types of renewable energy sources to generate electrical power, driven by concerns regarding energy security and the environmental impact resulting from carbon dioxide emissions [1,2]. Solar and wind energy, in particular, have gained widespread adoption worldwide [3,4]. For instance, photovoltaic solar panels produce variable low-voltage outputs ranging from 12–40 V, which are unsuitable for applications requiring high DC supply voltage or household appliances [5,6]. Therefore, DC-DC converters are employed to step up very low input voltage to high output DC voltage for various applications such as streetlights, motor drives, micro-grid systems, uninterruptible power supplies, fuel cells, and medical equipment [2,4–35]. Different converter topologies, including buck-boost, boost, and Ćuk DC-DC converters, have been introduced to achieve high voltage gain. The choice of converter depends on the specific application requirements. However, these converters encounter challenges when targeting ultra-high voltage gain. Key issues include a lower count of inductors and capacitors, reduced efficiency at high voltage gain with extremely high duty ratios, high voltage stress on power switches and diodes (equivalent to the output voltage), and increased current stress on power devices as the load current rises. In this context, current stress refers to the magnitude of current flowing through the switch during the on state and is influenced by the paths traversed by the current through the MOSFET. Moreover,

these converters suffer from elevated switching and conduction losses when the duty cycle exceeds 0.9, as well as low power density [20]. Although boost converter topologies can achieve voltage step-up with high voltage gains of up to 10, their efficiency diminishes at high duty cycles [10,36]. Several researchers have proposed new technologies for DC-DC converters to attain high voltage gain in renewable energy systems (RESs).

A modified DC-DC converter, interleaved with a boost converter based on soft switching between the main and auxiliary power MOSFETs, has been proposed to achieve high voltage gain and solve the inrush current problem [1]. Another modification to the boost converter using a hybrid inductor capacitor has been proposed in [8], and another with a switch capacitor (SC), switch inductor (SL), and voltage multiplier (VM) in [31]. A cascaded conventional SEPIC with a boost converter has been proposed in [24,26], and a modified SEPIC based on an interleaved buck-boost converter in [11]. A modified converter with multi-input has been proposed in [9,25]. These converters, as mentioned, are modified to achieve high voltage gain with high power density. However, they have low voltage gain and a high number of inductors and capacitors. Additionally, they require a high number of power diodes and power MOSFETs to step up the low input to a high output voltage. Furthermore, the power MOSFETs and diodes in these converters experience high voltage stress. Complex control was also required in [8] to achieve high voltage gain. Moreover, these converters achieve high voltage gain with an extremely high duty cycle, resulting in high switching and conduction losses, as well as low performance and efficiency. Other topologies have been proposed, such as a modified DC-DC converter with coupled inductors and a voltage multiplier network (VMN) [3], and a conventional SEPIC with a coupled inductor and VM [23]. These converters have demonstrated high voltage gain, but they feature a high number of passive and active elements, as well as diodes. The large number of components leads to a high parasitic resistance of inductors and capacitors, resulting in reduced efficiency. Additionally, these converters operate at a low switching frequency, which necessitates the use of large values of inductors and capacitors. Moreover, the internal resistance of the power MOSFET (Ron) is high, further diminishing the voltage gain of the system. Furthermore, a major issue with coupled inductor converters is the occurrence of high spike voltages in the off state of the power switch due to the inductance with parasitic capacitance of the power MOSFET switch [36]. To address this problem, a clamped circuit can be added to the power switch to prevent the occurrence of high spike voltages due to the coupled inductor [2,37]. However, adding more components to the circuit increases costs and reduces efficiency due to parasitism. In addition, the system will be heavy and large. Another problem is the pulsating input current at a low duty cycle, which makes these converters unsuitable for RES applications.

Other researchers have developed DC-DC converters to achieve high voltage gain using non-isolated coupled inductors. In [12], a double power switch converter with double switch inductor (SL) was utilized. Additionally, in [13], a modified converter incorporating a hybrid capacitor and inductor, and in [14], a DC-DC converter employing the voltage lift technique, are proposed. Furthermore, in [15,16], a DC-DC converter based on SC with zero voltage switching, and in [10,17], a modified buck-boost converter featuring a single switch and pulsating input current, are described. However, these converters require an extremely high duty ratio to achieve a high voltage gain ratio. This implies high switching and conduction losses, low efficiency, low power density, and high voltage stress on the power switches, diodes, inductors, and capacitors. Additionally, these converters exhibit a high inductor count with low switching frequencies, which results in high parasitic resistance, diminished performance, and efficiency. A modified boost converter with dual power switch was proposed in [18], a modified boost converter with a single switch, multiplier capacitor, and SL in [19,22], and a multi-input converter with multiple capacitors as hybrid energy storage in [21], with the goal of attaining high voltage gain for RESs. A buck-boost converter with SC SL was developed in [27], a pair of cascaded conventional boost converters with dual switches in [28] and with a single switch in [30], and a modified buck interleaved with SEPIC based on (SC) (SL) with two input sources in [29]. These converters

achieve high voltage gain; however, they have a large number of power switches and diodes, which has a high impact on system efficiency and performance. In addition, the voltage and current stress on the power switch is high. Furthermore, the gate control circuit is large and complex to implement.

In this paper, a new single-cell hybrid switched inductor-capacitor DC-DC converter is proposed to demonstrate the verification of ultra-high voltage gain in renewable energy applications (REA). The proposed modification involves incorporating a single cell of a hybrid inductor, along with a diode and double capacitors, interleaved with the main switch. This integration enables the proposed converter to attain a high voltage gain while ensuring that the input current does not experience zero pulsations at very low duty cycles. Additionally, the current stress on the main switch will reduce as the duty ratio increases, as shown in Figure 11f when the single-cell capacitor $C_1$ charge becomes zero. Furthermore, the converter employs a modified hybrid switch inductor in dual boosting mode, working with a single switch. This arrangement allows for the realization of an ultra-high voltage gain. Moreover, one of the passive components, $L_2$, will be open circuit, and $D_1$ is working at zero current switching, which means a reduction in the total power loss of the converter.

## 2. Structure and Operation of the Proposed DC–DC Converter

The DC-DC converter is modified to achieve a high voltage gain by stepping up a low input voltage range of 20–40 V to an output voltage of 200 V. The proposed converter is based on a modified hybrid switched inductor-capacitor configuration to verify high voltage gain. Figure 1a shows the basic connection of a switched inductor, while Figure 1b illustrates the hybrid connection of a switched inductor and a capacitor. Consequently, the proposed converter is designed for Renewable Energy Sources (RES). In Figure 1c, the connection of the proposed converter with PV Panels and Battery is shown for Energy Saving Mode Applications. The structure of the proposed converter includes four inductors, four capacitors, three diodes, and two power switches, as depicted in Figure 1d. The main advantages of the proposed converter are that it utilizes non-coupled inductors and is transformerless. It employs a high switching frequency to reduce the sizes of inductors and capacitors, thereby increasing its efficiency. The proposed converter has a simple structure with a straightforward control circuit. The new single cell of an inductor and capacitor, which is interleaved with the main switch, helps avoid pulsating input current at low duty cycles and minimizes voltage stress on the power switches, diodes, and inductors. Additionally, the current stress on the main switch is reduced when the duty cycle increases. Furthermore, the proposed converter achieves an ultra-high voltage gain compared to previous DC-DC converters. In terms of the power switch PWM generator, it uses a simple design where both MOSFETs turn on and off simultaneously. The proposed converter can operate in DCM under two cases. Firstly, it can function in DCM Case 1 (DCMC1) at a low duty cycle and maximum input voltage, with a duty ratio below 50%. Secondly, the proposed converter can operate in DCM Case 2 (DCMC2) when the input voltage decreases during the day at a high load current, with a duty ratio above 50%. Additionally, the converter can also operate in CCM when the load current increases and the duty ratio exceeds 70%. These scenarios are illustrated in Figure 4a,b, which demonstrates the dynamic performance of the proposed converter.

### 2.1. Proposed Converter Operation DCMC1

The proposed converter can operate in DCMC1 in four states of operation during one cycle when the input voltage is at its maximum, with a low duty cycle, and at light load. The waveform of this mode of operation is shown in Figure 2a. The four states of this operational mode are listed below:

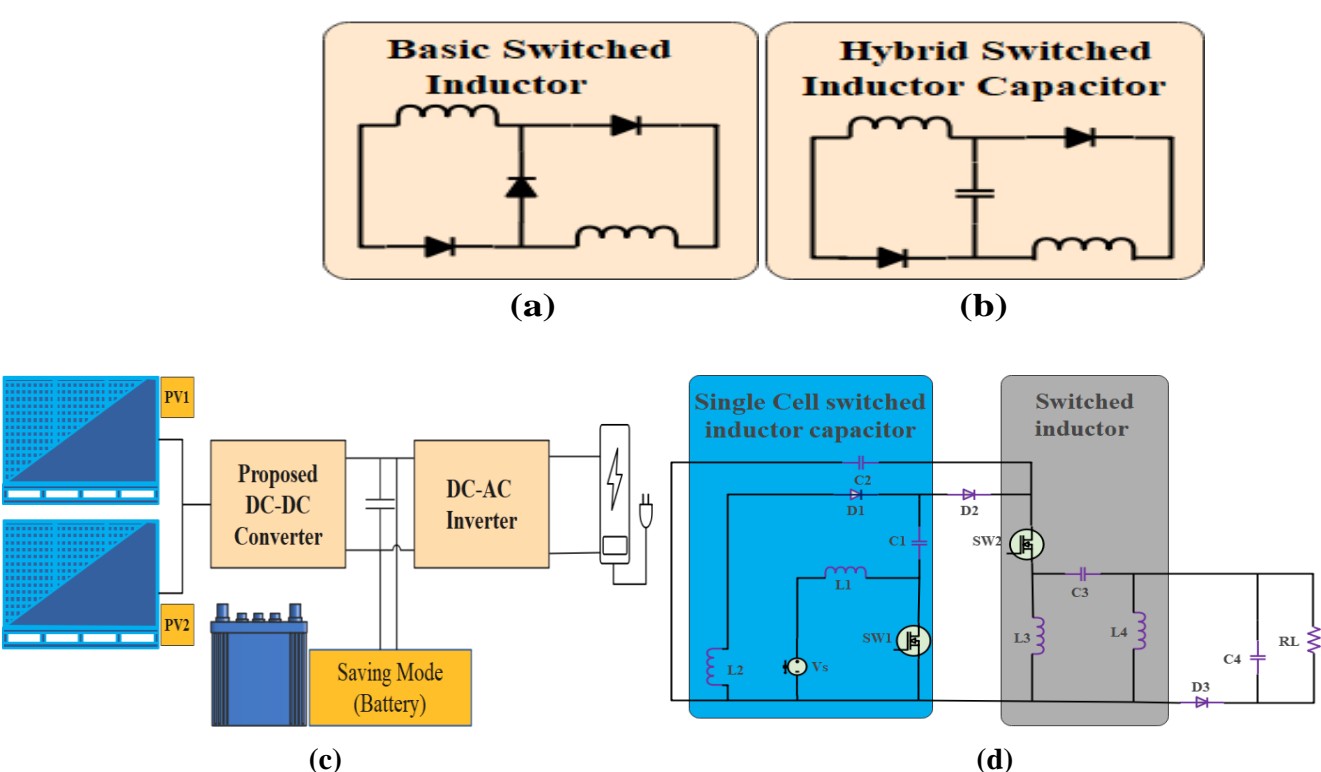

**Figure 1.** (**a**) Basis circuit of the switched inductor (**b**) Hybrid switched inductor-capacitor connection (**c**) Schematic Diagram: Connection of the proposed converter with PV panels and battery for energy saving mode applications (**d**) The Proposed DC-DC Converter with single-cell switched inductor capacitor interleaved with modified switched inductor.

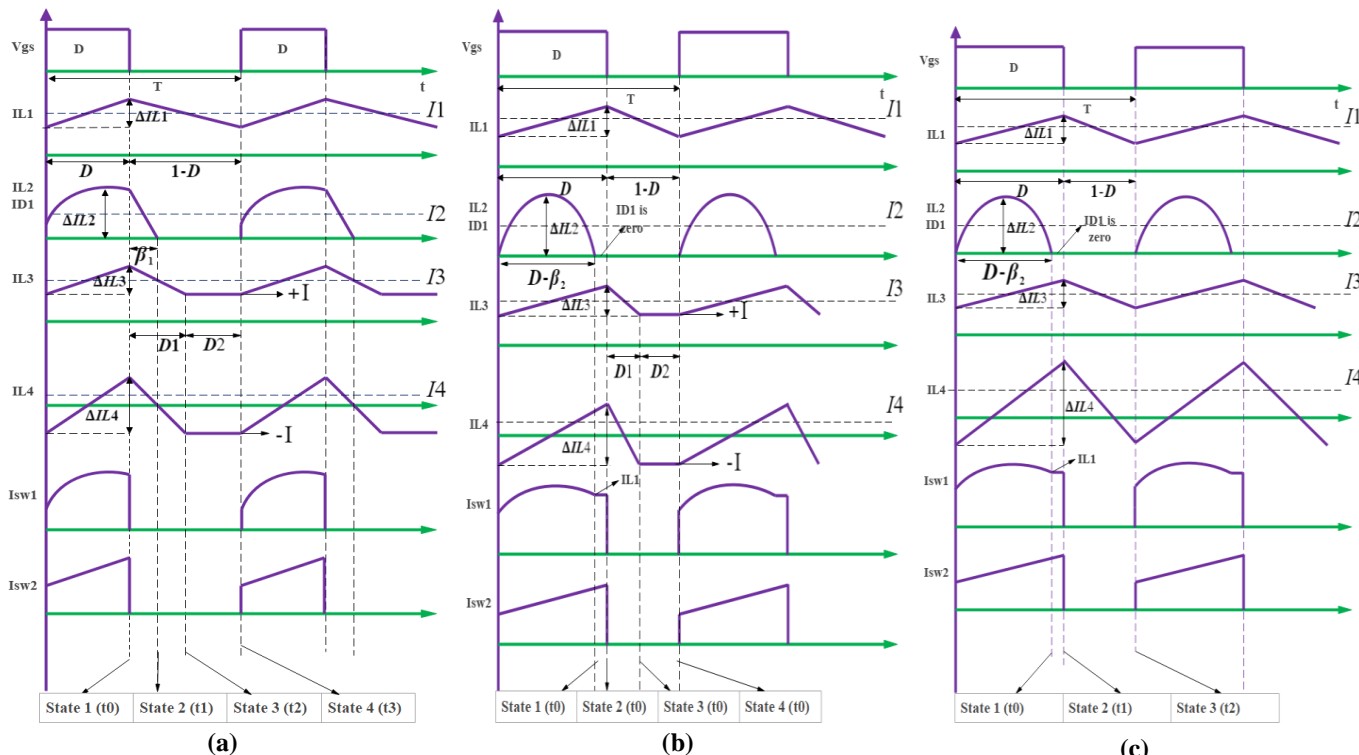

**Figure 2.** (**a**) Proposed Converter waveforms at DCMC1, (**b**) Proposed Converter waveforms at DCMC2, (**c**) Proposed Converter waveforms at CCM.

State 1: [0–$t_0$]: In this mode, the two power MOSFETs ($Sw_1$ and $Sw_2$) are in the on state, and a PWM generator provides a high source-to-gate voltage to turn both MOSFETs on and keep diodes $D_2$ and $D_3$ off. During this mode, $L_1$ charges with energy from the input source, which is connected in series with it. $L_2$ begins to charge from $C_1$, and $C_1$ discharges the energy stored in $L_2$ through $Sw_1$. $D_1$ is on during this mode, and $C_2$ stores a significant amount of energy, which is used to charge $L_3$ through $Sw_2$. At the same time, $L_4$ starts charging from $C_3$. $C_4$ supplies power to the load and forms the current path for this mode, as shown in Figure 2b.

The voltage equations of this mode for inductors, capacitors, diodes, and MOSFETs are as follows:

$$\left.\begin{aligned}
VL_1 &= Vs \\
VL_2 &= Vc_1 \\
VL_3 &= Vc_2 \\
VL_4 &= Vc_2 - Vc_3 \\
Vc_4 &= Vo
\end{aligned}\right\} \tag{1}$$

The current equations of this mode for inductors, capacitors, diodes, and MOSFETs are as follows:

$$iL_1 + iL_2 = ISW_1 \tag{2}$$

$$ISW_2 = Ic_2 = iL_3 + iL_4 \tag{3}$$

$$ID_1 = iL_2 = IC_1 \tag{4}$$

$$IC_4 = Io \tag{5}$$

$$\left.\begin{aligned}
iL_1 &= \frac{Vs}{L_1} \\
iL_2 &= \frac{Vc_1}{L_2} \\
iL_3 &= \frac{Vc_2}{L_3} \\
Io &= \frac{Vo}{RL}
\end{aligned}\right\} \tag{6}$$

where VL represents the voltage across the inductor, Vc denotes the voltage across the capacitor, Isw represents the current through the power MOSFET switch during the on state, Vo signifies the output voltage, iL represents the current through the inductor, Io represents the output current, RL represents the resistive load, ID is the current through the power diode, and Ic denotes the current through the capacitor.

State 2: [$t_0$–$t_1$]: In this mode, the two power MOSFETs are in the off state, and both diodes $D_2$ and $D_3$ in the on state. $L_1$ discharges energy to $C_1$ and charges $C_2$. $L_2$ starts discharging its energy to $C_2$ through $D_1$ and remains in the on state. $C_2$ stores a large amount of energy from $L_1$ and $L_2$. $L_3$ and $L_4$ start discharging their energy to $C_4$, which supplies high power to the load and forms the current path of this mode, as shown in Figure 3c.

The voltage equations of this mode for inductors, capacitors, diodes, and MOSFETs are as follows:

$$\left.\begin{aligned}
VL_1 &= Vs - Vc_1 - Vc_2 \\
VL_2 &= -Vc_2 \\
VL_3 &= Vc_3 + VL_4 \\
VL_4 &= -Vc_4 \\
Vc_4 &= V_o
\end{aligned}\right\} \tag{7}$$

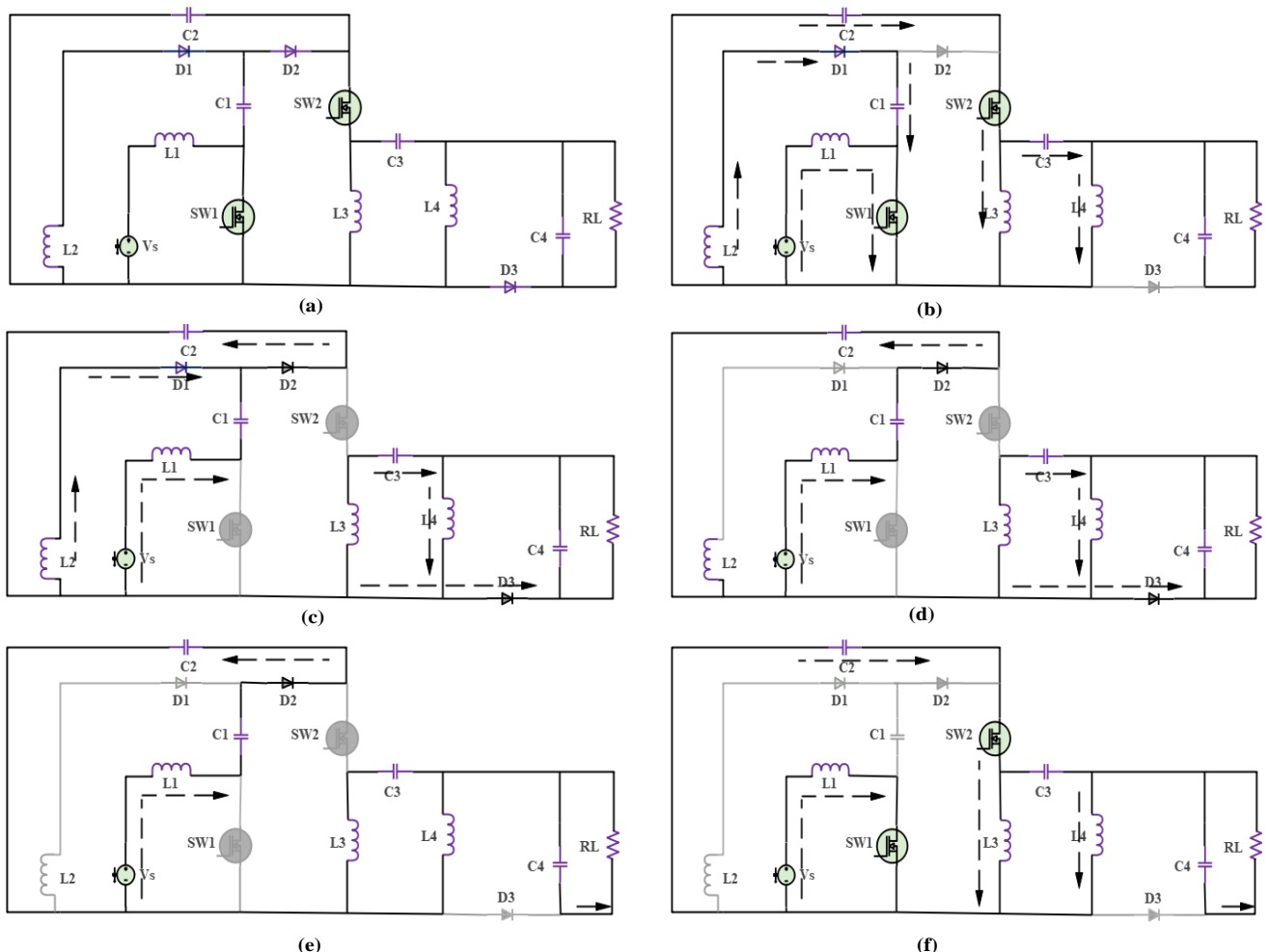

**Figure 3.** (**a**) proposed DC-DC converter (**b**) state 1 mode DCMC1, state 1 mode DCMC2, state 1 mode CCM (**c**) state 2 mode DCMC1 (**d**) state 3 DCMC1, state 3 DCMC2, state 3 CCM (**e**) state 4 DCMC1 state 4 DCMC2 (**f**) state 2 mode DCMC2, state 2 mode CCM.

The current equations of this mode for inductors, capacitors, diodes, and MOSFETs are as follows:

$$\left.\begin{array}{l} iL_1 = \frac{VS}{L_1} - \frac{Vc1}{L_1} - \frac{Vc2}{L_1} \\ iL_2 = \frac{-Vc_2}{L_2} \\ iL_3 = \frac{Vc_3 + VL_4}{L_3} \\ Io = \frac{Vo}{RL} \end{array}\right\} \tag{8}$$

$$iL_1 + iL_2 = ID_2 \tag{9}$$

$$iL_3 + iL_4 = ID_3 \tag{10}$$

$$ID_1 = iL_2 \tag{11}$$

$$Ic_4 = Io = iL_3 + iL_4 \tag{12}$$

$$iL_1 = Ic_1 = Ii \tag{13}$$

where Ii is the input current, which is equal to $iL_1$.

State 3: [$t_1$–$t_2$]: In this mode, the two power MOSFETs are still in the off state, and both diodes $D_2$ and $D_3$ are still in the on state. $L_1$ continues discharging energy to $C_1$ and charges $C_2$. $L_2$ reaches zero charge and $iL_2$ and $ID_1$ are zero. $C_2$ continues receiving energy from only $L_1$, and $D_1$ is now in the off state in this mode. $L_3$ and $L_4$ continue discharging their energy to $C_4$, which supplies high power to the load and forms the current path of this mode, as shown in Figure 3d.

The current equations of this mode are as follows:

$$iL_1 = ID_2 \tag{14}$$

$$ID_1 = iL_2 = 0 \tag{15}$$

$$Ic_2 = iL_1 \tag{16}$$

State 4: [$t_2$–$t_3$]: In this mode, the two power MOSFETs are still off, the PWM generator gives zero gate-to-source voltage to keep them in the off state, and only $D_2$ is still on. $D_3$ is now changed to the off state in this mode. $L_1$ continues discharging energy to charge $C_1$ and $C_2$. $C_2$ stores a large amount of energy for the next pulse to supply it to the load. $L_2$ has zero charge and $iL_2$ and $ID_1$ are zero, and $D_1$ is still off in this mode. $L_3$ and $L_4$ will have the same current values but in opposite directions: $iL_3 = -iL_4$. $C_4$ supplies high power to the load and forms the current path of this mode, as shown in Figure 3e.

Figure 4a,b illustrate the dynamic performance of the proposed converter when operating in DCM and CCM, respectively. It is evident from the figures that the proposed converter operates in DCM for duty cycles below 70%. However, for duty cycles above 70%, the converter can operate in CCM based on the boundary condition specified in Equation (33).

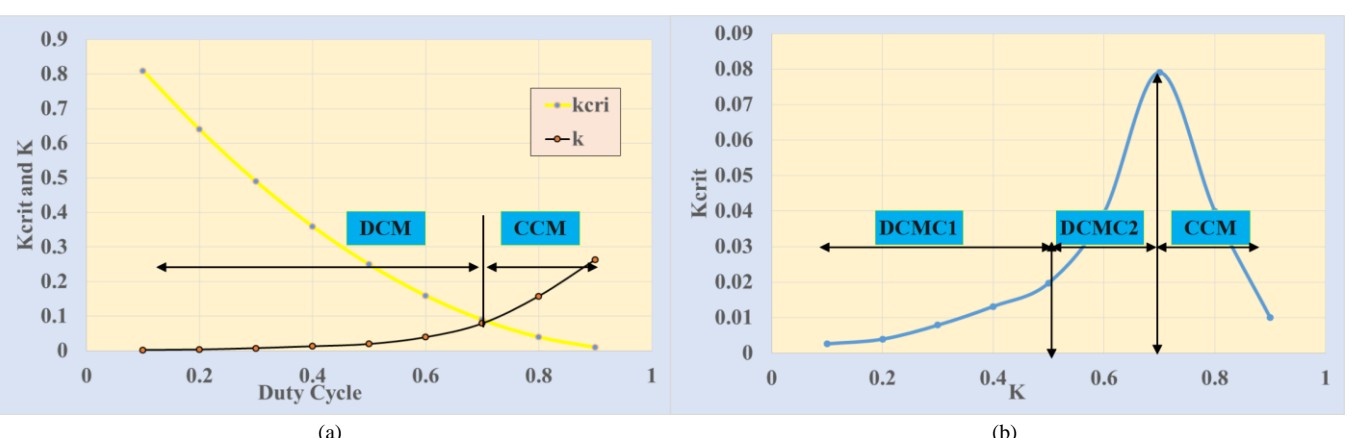

**Figure 4.** Dynamic performance of the proposed converter (**a**) Loading factor K and Kcrit Vs duty cycle, (**b**) Kcrit Vs K.

### 2.2. Proposed Converter Operation in DCMC2

The proposed converter can operate in DCMC2 when the input voltage is reduced to the minimum value during the day, while the load current is high. In this mode, the input current is still in CCM, and $L_2$ will be in resonant mode with $C_1$, while $L_3$ and $L_4$ will continue to operate in DCM. Therefore, this mode has four states of operation, as shown in Figure 2b: the proposed converter waveforms at DCMC2.

State 1: [$0$–$t_0$]: In this mode, the two MOSFETs $Sw_1$ and $Sw_2$ are in the on state, and both diodes $D_2$ and $D_3$ in their off states. $L_1$ starts charging energy from the input source, which is connected in series with it in CCM and never reaches zero. Inductor $L_2$ starts charging from $C_1$, and $C_1$ will discharge energy to $L_2$ through $Sw_1$ until $(D - \beta_2)$, the charge across $C_1$ will be zero, and the current through $Sw_1$ will come only from $IL_1$ after

this time, as shown in Figure 11f. $D_1$ is on in this mode, and $C_2$ stores a large amount of energy, which charges $L_3$ through $Sw_2$, and $L_4$ starts charging from $C_3$. $C_4$ supplies power to the load and forms the current path of this mode, as shown in Figure 3b. The current and voltage equations of this mode are the same as for State 1 of DCM Case 1. Only the current through $Sw_1$ is shown below:

$$\left.\begin{array}{l} Isw_1 = (iL_1 + Ic_1) \text{ from } 0 < t < (D - \beta_2) \\ Isw_1 = iL_1 \text{ only from } (D - \beta_2) < t < D \end{array}\right\} \tag{17}$$

$$iL_2 = \frac{Vc_1}{L_2}(D - \beta_2) \tag{18}$$

State 2: [$t_0$–$t_1$]: In this mode, the two power Mosfets are still in the on state. The PWM generator continues to provide a high gate-to-source voltage to keep them in the on state, and both diodes $D_2$ and $D_3$ remain off. Meanwhile, $L_1$ continues to charge energy from the input source, while $L_2$ reaches zero charge due to the charge across $C_1$ being zero. The current through $Sw_1$ comes solely from $IL_1$, reducing the current stress through Sw1. $D_1$ operates with zero current switching during this mode, meaning that $L_2$ will be open circuit during the time period (($D - \beta_2$) < t < D). $L_3$ and $L_4$ continue to charge from $C_2$ and $C_3$, respectively. C4 continues to supply power to the load, and the current path of this mode is shown in Figure 3f. The current equations remain the same as in the previous state 1.

State 3: [$t_1$–$t_2$]: in this mode, the two power MOSFETs are off, and both diodes $D_2$ and $D_3$ are now on. $L_1$ now starts discharging energy to $C_1$ and charges $C_2$. $L_2$ does not charge, and $iL_2$ and $ID_1$ are zero. $L_2$ is open circuit during this mode. $C_2$ receives a large amount of energy from $L_1$. $L_3$ and $L_4$ also start discharging their energy to $C_4$, which supplies high power to the load and forms the current path of this mode, as shown in Figure 3d.

State 4: [$t_2$–$t_3$]: In this mode, the two power MOSFETs are still off, and only $D_2$ is still on. $D_3$ is changed to the off state in this mode. $L_1$ continues discharging energy to charge $C_2$, and $C_2$ will have a large amount of energy for the next pulse to supply it to the load. $L_2$ remains open circuit in this mode. $C_2$ will continue receiving energy only from $L_1$. $L_3$ and $L_4$ will have the same current but in opposite directions: $iL_3 = -iL_4$. $C_4$ will supply high power to the load and forms the current path of this mode, as shown in Figure 3e.

When the proposed converter operates in DCMC2, the capacitor $C_1$ is discharged to zero at ($D - \beta_2$), reducing the current stress across $Sw_1$. In this case, the current through $Sw_1$ flows only from $L_1$. Additionally, the voltage stress across $Sw_1$ and the power diodes is significantly reduced. Furthermore, the conduction loss of $Sw_1$ is significantly reduced when the stress current is reduced. In this mode, $L_2$ will be in an open circuit state, and $D_1$ will have very low voltage stress. Additionally, as mentioned above, $C_1$ is discharged to zero. This implies that the components of the proposed converter are reduced during this mode, leading to improved performance and efficiency.

*2.3. Proposed Converter Operation in CCM*

This operation mode occurs when the load current increases to a duty cycle of 70%, as depicted in Figure 4. The input current still operates in CCM, and $L_2$ enters a resonant mode with $C_1$, while $L_3$ and $L_4$ continue to operate in CCM. Additionally, the voltage gain in this mode will be increased. Thus, this mode consists of three states of operation, as illustrated in Figure 2c, which shows the proposed converter waveforms in CCM.

State 1: [0–$t_0$]: in this mode, same as state 1 DCMC2.

State 2: [$t_0$–$t_1$]: in this mode, same as state 2 DCMC2.

State 3: [$t_1$–$t_2$]: In this mode, the two power MOSFETs are off, and both diodes $D_2$ and $D_3$ are now on. $L_1$ will start discharging energy to $C_1$ and charge $C_2$, and $iL_2$ and $ID_1$ are zero. $C_2$ receives a large amount of energy from $L_1$. $L_3$ and $L_4$ will also start discharging their energy to $C_4$, and $L_3$ and $L_4$ will have the same current in CCM but in opposite directions: $iL_3 = -iL_4$. $C_4$ will supply high power to the load and forms the current path of this mode, as shown in Figure 3d.

### 3. Voltage Gain Calculations of the Proposed Converter

In this section, the voltage gain of the proposed converter is calculated when it operates in DCM and CCM.

*3.1. Voltage Gain of the Proposed Converter in DCMC1*

In this section, we analyze the voltage gain of the proposed converter while it operates in DCMC1. This particular mode is encountered when the input voltage reaches its maximum value during light load applications, with a duty cycle below 50%. Hence, the equations that describe the voltage gain under the DCMC1 operation of the proposed converter are presented below:

$$\frac{1}{Ts}\left(\int_0^{DTs}(Vs+Vc_1)dt + \int_D^{Ts}(Vs-Vc_1-Vc_2)dt + \int_{DTs}^{\beta_1 Ts}(-Vc_2)dt\right) = 0 \qquad (19)$$

$$\frac{1}{Ts}\left(\int_0^{DTs}(Vc_2)dt + \int_D^{D1Ts}(Vc_2-Vc_3)dt = 0 \qquad (20)$$

$$Vc_2 = \frac{Vs}{(1+\beta_1-D)} \qquad (21)$$

$$\beta_1 = \left(\frac{Vs}{Vc_2}+D-1\right) \qquad (22)$$

$$D1 = \frac{DVc_2}{Vo} \qquad (23)$$

From Equations (1) and (7), applying volt-second balance to $L_1$ and $L_2$ results in Equation (19). Solving it results in Equation (21). Applying volt-second balance to $L_3$ and $L_4$ from Equations (1) and (7) yields Equation (20). After solving Equation (20), by applying the fact that the average voltage across $C_3$ is almost zero during the steady state, Equation (23) can be obtained (D1) is the discharging time of $L_3$ and $L_4$, and ($\beta_1$) represents the discharging time of $L_2$, which can be found from Equation (22) and is a function of $(D, Vs, Vc_2)$.

$$\left.\begin{array}{l} <I2> = \frac{Vc_1}{2L_2}DTs(D+\beta1) \\ <I3> = \frac{Vc_2}{2L_3}DTs(D+D1)+I \\ <I4> = \frac{Vc_2}{2L_4}DTs(D+D1)-I \end{array}\right\} \qquad (24)$$

$$\left.\begin{array}{l} iL_{2peak} = \frac{VsDTs}{2(1-D)L_2} \\ iL_{3peak} = \frac{VsDTs}{(1-D)L_3} \\ iL_{4peak} = \frac{VsDTs}{(1-D)L_4} \end{array}\right\} \qquad (25)$$

$$I3+I4 = Io \qquad (26)$$

$$Io = \frac{VsDTsD1}{2Le(1+\beta_1-D)} \qquad (27)$$

To find the average current through $L_2$, $L_3$, and $L_4$, we can use Equation (24) to determine their respective averages. By adding the average currents $I_3$ and $I_4$ in Equation (24), we can calculate the average output current using Equation (27). The peak current of the inductors $L_2$, $L_3$, and $L_4$ can be found using Equation (25).

$$Vc_2 = \sqrt{\frac{2LeVo^2}{RLD^2Ts}} \qquad (28)$$

$$Le = \frac{L_3L_4}{L_3+L_4} \qquad (29)$$

$$Gv(DCMC1) = \frac{VsD^2TsRL}{2LeVo(1 + \beta_1 - D)^2} \tag{30}$$

$$Gv(DCMC1) = \frac{D}{(1 + \beta_1 - D)\sqrt{K}} \tag{31}$$

$$k_{crit} = \frac{(1 - D)^4}{(1 + \beta_1 - D)^2} \tag{32}$$

$$k_{crit} = \begin{cases} If \text{ Kcrit} > \text{ K Proposed Converter work in DCM} \\ If \text{ Kcrit} < \text{ K Proposed Converter work in CCM} \end{cases} \tag{33}$$

The voltage across capacitor $C_2$ can be obtained using Equation (28), which is a function of RL, Vo, D, and Le when the converter operates in DCMC1. Here, Le represents the inductor equivalent of $L_3$ and $L_4$, which can be found in Equation (29). Equation (30) provides the voltage gain of the proposed converter at DCMC1. Equation (31) expresses the voltage gain equation of the proposed converter in DCMC1 as a function of the load loss factor (K). The boundary condition of the proposed converter, specifically the critical load loss factor ($K_{crit}$), can be determined using Equations (32) and (33) by applying the values of ($\beta_1$) in Equation (32).

### 3.2. Voltage Gain of Proposed Converter in DCMC2

In this section, the voltage gain of the proposed converter can be adjusted by varying the input voltage and the load resistance. This mode occurs when the input voltage source reaches its minimum value at a duty cycle above 50%, while the proposed converter provides a high load current. Consequently, the voltage gain of the proposed converter in the DCMC2 equations is presented below.

$$\left.\begin{array}{l} \frac{1}{Ts}(\int_0^{DTs}(VS)dt + \int_0^{(D-\beta_2)Ts}(Vc1)dt + \int_{DTs}^{Ts}(Vs - Vc1 - Vc2)dt)) = 0 \\ \frac{1}{Ts}(\int_0^{DTs}(Vc2)dt + \int_D^{D1Ts}(Vc2 - Vc3)dt = 0 \end{array}\right\} \tag{34}$$

$$D(Vin) + (1 - D)(Vin - Vc1 - Vc2) + (D - \beta_2)(Vc1) = 0 \tag{35}$$

By applying voltage second balance on $L_1$, $L_2$, $L_3$ and $L_4$, we obtain Equation (34) as a result in Equation (35). From Equation (36), we determine the value of $Vc_1$, which is almost equal to the difference between the input voltage and the voltage across capacitor $C_2$. $Vc_2$ can be found in Equation (37). The values of ($\beta_2$) is adjusted and can be found in Equation (38), depending on the values of $C_1$ and $L_2$. After using Equation (34) and concluding with Equation (40), we obtain the voltage gain of the proposed converter when operating in DCMC2.

$$Vc_1 = Vs - Vc_2 \tag{36}$$

$$Vc_2 = \frac{Vs}{(1 - D)} \tag{37}$$

$$\beta_2 = D - \frac{\sqrt{C_1L_2}}{Ts} \tag{38}$$

$$D1 = \frac{DVc_2}{Vo} \tag{39}$$

$$Gv(DCMC2) = \frac{VsD^2TsRL}{2LeVo(1 - D)^2} \tag{40}$$

### 3.3. Voltage Gain of Proposed Converter in CCM

The voltage gain of the proposed converter is derived in CCM as shown in the equations below:

$$\frac{1}{Ts}\left(\int_0^{DTs}(Vc_2)dt + \int_D^{Ts}(Vc_2 - Vc_3)dt = 0\right) \tag{41}$$

$$Vo = \frac{Vc2D}{(1-D)} \tag{42}$$

$$Gv = \frac{Vo}{Vs} = \frac{D}{(1-D)^2} \tag{43}$$

By using the same Equations (34) and (35), by applying volt-second balance to $L_1$ and $L_2$ and, to $L_3$ and $L_4$, we obtain Equation (41) at different times. The result can be found in (42) and (43). The voltage gain of the proposed converter in CCM is given by Equation (43).

The voltage gain equations of the proposed converter in DCM and CCM demonstrate that it has a higher voltage transfer gain compared to previous DC-DC converters. Additionally, two passive components, $L_2$ and $C_1$, become open circuits as depicted in Figure 3f. Moreover, $D_1$ operates at Zero Current Switching and experiences low voltage stress, thereby reducing the total power loss of the converter. This reduction occurs because $L_2$ reaches zero charging when the charge across $C_1$ becomes zero. The current flowing through $Sw_1$ is solely sourced from $iL_1$.

### 4. Voltage across Power Diodes, MOSFETs and Capacitor

The proposed converter has two power switches, three power diodes, and four capacitors. Therefore, in this section, the voltage stress across power MOSFETs is calculated. Furthermore, the voltages across the power diodes and capacitors of the proposed converter are also calculated.

$$V_{D_1} = \frac{Vs}{(1-D)} \tag{44}$$

$$V_{D_2} = \frac{Vs}{(1-D)} \tag{45}$$

$$V_{D_3} = \frac{VsD}{(1-D)^2} \tag{46}$$

In order to find the voltage stress across the power diodes in the proposed converter, we can use Equation (44) to determine the voltage stress across $D_1$, which is a very small value. Equation (45) can be used to obtain the voltage stress across $D_2$, while Equation (46) provides the means to find the voltage stress across $D_3$. It can be observed that the voltage across the power diodes is very small and depends on the input voltage, which ranges from 20 V to 40 V.

$$Vsw_1 = \frac{VsD}{(1-D)} \tag{47}$$

$$Vsw_2 = \frac{Vs}{(1-D)}, average\ voltage \tag{48}$$

To determine the voltage stress across the Power MOSFETs, we can use Equation (47) to find the voltage across MOSFET $Sw_1$. This voltage is also very small and depends on the input voltage, which varies from 20 V to 40 V. Additionally, Equation (48) allow us to calculate the average voltage stress across $Sw_2$. During the time period $D < t < D1$, the voltage across $Sw_2$ is equal to the output voltage. During the time period $D1 < t < Ts$, it is equal to the average voltage across $C_2$.

$$Vc_1 = \frac{Vs}{(1-D)} - \sqrt{\frac{2LeVo^2}{RLD^2Ts}}\ (D < 0.5) \tag{49}$$

$$Vc_2 = \frac{Vs}{(1-D)} \tag{50}$$

$$Vc4 = Vo \tag{51}$$

To determine the voltage across capacitors, we can refer to Equation (49) for the voltage across $C_1$ and Equation (50) for the voltage across $C_2$. Additionally, Equation (51) provides the means to find the voltage across the filter capacitor, which is equal to the output voltage. By reducing the voltage stress on all power diodes and power MOSFETs, the proposed converter experiences decreased losses, leading to improved efficiency.

## 5. Features Components of Proposed Converter

In this section, we discuss the main components of 200 W prototype of proposed converter, which include inductors, capacitors, power diodes, power Mosfet, and gate drive circuit. Therefore, these components are crucial for designing and verifying the high voltage gain of the proposed converter. The suggested converter has four small values of inductors, four small values of capacitors. Table 1 provides the parameters of the prototype design for the proposed converter.

**Table 1.** Prototype parameters design for the proposed converter.

| | |
|---|---|
| SiC MOSFET | 650 V, 40 A. 35 mΩ |
| SiC Schottcky diode | 1200 V 40 A |
| $L_1$ | 100 uh 2.9 mΩ |
| $L_2$ | 3 uh 1.5 mΩ |
| $L_3$ | 100 uh 2.9 mΩ |
| $L_4$ | 15 uh 2.2 mΩ |
| $C_1$ | 1 uF 100 V |
| $C_2$ | 200 uF 100 V |
| $C_3$ | 2 uF 500 V |
| $C_4$ | 100 uF 500 V |
| Vs | 20–40 V |
| Vo | 200 V |
| Power | 200 w |
| Duty Cycle (D) | 0.45 |
| Fs Switching Frequency | 150 KHZ |
| Inductor size ($L_1 = L_3 = L_4$) | (L/2.85 cm·W/2.75 cm·H/2.5 cm) |
| Inductor size $L_2$ | (L/1.85 cm·W/1.5 cm·H/2 cm) |

It can be observed from Table 1 that the values of the inductors become significantly smaller when a high switching frequency is employed. Furthermore, the internal resistance of the inductors is also minimized, resulting in reduced total power losses for the proposed converter. The sizes of the inductors are provided in Table 1. It is evident that by utilizing a high switching frequency, the overall dimensions of the proposed converter are substantially reduced. The inductor specifications used in the proposed converter involve a ferrite core with flat wire to minimize internal resistance. To design the inductors of the proposed converter, $L_1$ can be designed using Equation (52), $C_1$ and $L_2$ can be designed using Equation (53). Inductor $L_3$ and $L_4$ can be found using Equation (54). $C_2$, $C_3$, and $C_4$ can be determined using Equations (55), (56), and (57), respectively.

$$L_1 \geq \frac{VsD}{Fs\Delta iL1} \tag{52}$$

$$\left.\begin{array}{l} L_2 = \frac{4\pi^2 Fs^3 RL(1-D)\Delta Vc_1}{DVo} \\ C_1 = \frac{VoD}{\Delta Vc_1 FsRL(1-D)} \end{array}\right\} \tag{53}$$

$$\left.\begin{array}{l} L_3 \geq \frac{VsD}{Fs\Delta iL_3(1-D)} \\ L_4 \geq \frac{8VsD}{Fs\Delta iL_4(1-D)} \end{array}\right\} \tag{54}$$

$$C_2 = \frac{VoD}{\Delta Vc2RLFs(1-D)} \tag{55}$$

$$C_3 = \frac{6VoD}{FsRL} \tag{56}$$

$$C_4 = \frac{VoD}{\Delta Vc_4 FsRL} \tag{57}$$

## 6. Comparison the Proposed Converter with Previous DC DC Converters

In this section, the proposed converter is compared to previous DC-DC converters. Both the previous works and the proposed converter are simulated in Matlab Simulink under the same conditions. From Figure 5, it can be observed that the voltage gain of the proposed converter is higher than that of the previous converters. A higher gain at a low duty cycle implies lower conduction losses, lower switching losses, higher efficiency, and a reduced number of inductors and capacitors. As shown in Figure 6a, the power MOSFET in the proposed converter experiences lower voltage stress compared to the power MOSFET in the previous converters. However, the voltage stress across the power MOSFET in the proposed converter slightly increases as the voltage gain increases. Regarding the voltage across the diodes, Figure 6b demonstrates that the voltage stress across the diode in the proposed converter is lower than in previous DC-DC converters.

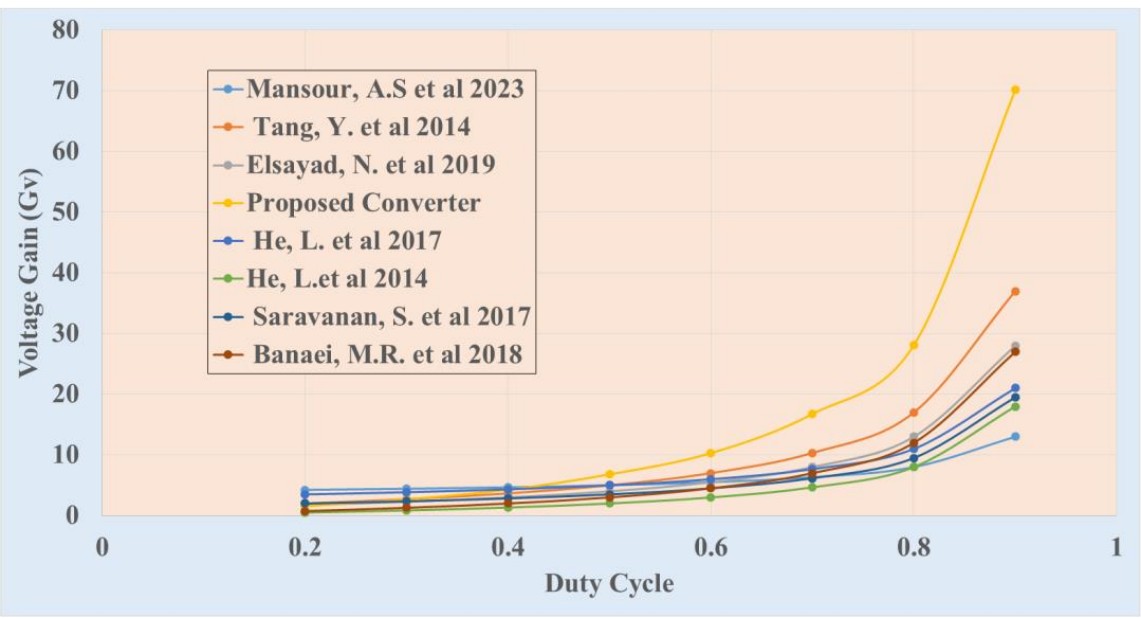

**Figure 5.** Voltage gain vs. Duty cycle [9,13,15,19,21,31,36].

The proposed DC-DC converter is compared with the previous DC-DC converters in Table 2. It can be seen that the proposed converter operates at a higher switching frequency than previous DC-DC converters. A low switching frequency requires high values of inductors and capacitors with high internal resistance, resulting in high switching and

conduction losses, particularly at high duty cycles. Furthermore, high values of passive components result in high weight, high cost, and large size.

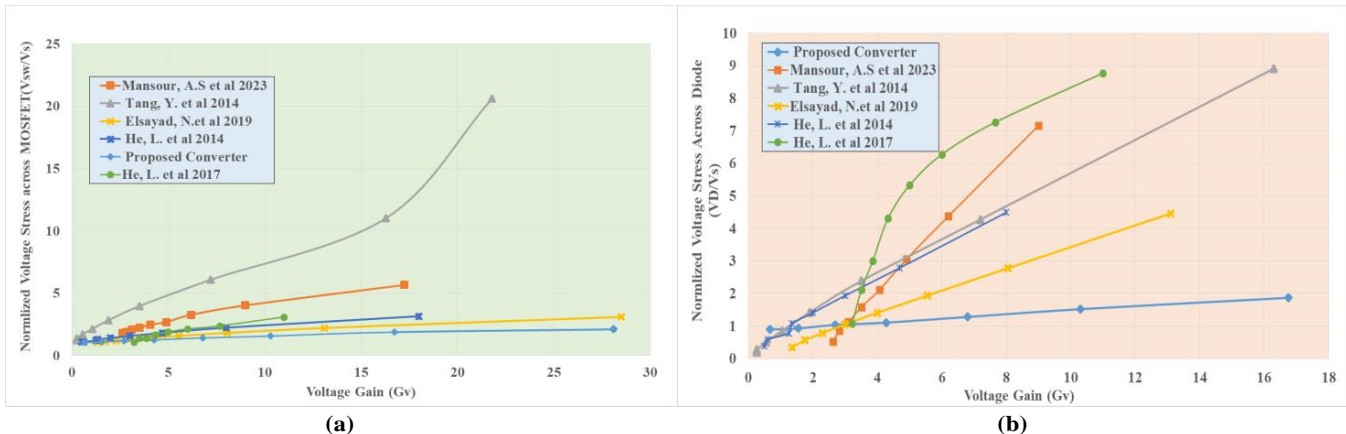

**(a)**          **(b)**

**Figure 6.** (**a**) Voltage stress across Power MOSFETs Vs. voltage gain, (**b**) Voltage stress across Diode Vs. Voltage Gain [9,13,15,19,36].

**Table 2.** Comparison between the Proposed Converter and Previous DC DC Converters.

| Items | Proposed Converter | Ref [36] | Ref [9] | Ref [13] | Ref [15] | Ref [19] | Ref [21] | Ref [4] | Ref [31] |
|---|---|---|---|---|---|---|---|---|---|
| Switching Frequency | 150 | 5 | 50 | 100 | 50 | 66 | 24 | 45 | 30 |
| Vs | 20–40 | 24 | 20–40 | 50 v | 15 v | 24 | 20 | 20 | 25 v |
| Vo | 200 v | 107 v | 200 v | 400 v | 90 v | 180 | 300 v | Out1 = 155 | 110 v |
| Inductors | 4 | 2 | 4 | 3 | 3 | 3 | 2 | 2 | 4 |
| capacitors | 4 | 3 | 1 | 5 | 4 | 3 | 4 | 8 | 6 |
| Diodes | 3 | 4 | 7 | 4 | 5 | 2 | 4 | 7 | 3 |
| switches | 2 | 1 | 2 | 1 | 2 | 1 | 1 | 1 | 1 |
| Duty cycle | 45% | 64% | 70% | 72% | 50% | 88% | 77% | 77% | 60% |
| Power (w) | 200 | 52 | 200 | 200 | 200 | 100 | 250 | 250 | 110 |
| Efficiency | 96.5% | 91.2% | 90% | 94.5% | 95% | 94% | 93.5% | 90% | 94.5% |
| Input Current | No Pulsating | Pulsating | Pulsating | Pulsating | No Pulsating | Pulsating | No Pulsating | Pulsating | Pulsating |
| Gain | $\frac{D}{(1-D)^2}$ | $\frac{(4-3D)}{(1-D)}$ | $\frac{(1+3D)}{(1-D)}$ | $\frac{(1+2D)}{(1-D)}$ | $\frac{(3-D)}{(1-D)}$ | $\frac{(2D)}{(1-D)}$ | $\frac{(3+D)}{2(1-D)}$ | $\frac{3}{(1-2D)}$ | $\frac{(3D)}{(1-D)}$ |

The proposed converter can step up a low input voltage at a duty cycle of 45% with a load of 200 W, but other converters in Table 2 can step up a low voltage at a high duty ratio. A high diode count leads to high internal resistance and high forward voltage (Vf), limiting the voltage gain and increasing the converter losses. In addition, a high diode count increases the reverse recovery time, which also affects the system's performance. In addition, the proposed converter has zero pulsating input current at low and high duty cycles compared to the input current in the previous DC-DC converter. In terms of the voltage gain equations of the proposed and previous DC-DC converters, as shown in Table 2, the proposed converter can achieve a higher voltage gain than the previous DC-DC converters listed in Table 2. Moreover, this means that the proposed converter is more efficient for applications that require high DC voltage gain at different loads, with more flexibility in the duty ratio. It achieves an efficiency of 96.5%. Additionally, the proposed converter is more suitable for RES.

## 7. Control Strategy of the Proposed Converter

The suggested controller, as shown in Figure 7, uses dual PI controllers to enable the proposed converter to operate with high performance. The first PI controller, known as the inner loop controller, is designed to control the load current. The second PI controller, called the outer loop controller, is responsible for controlling the output voltage of the proposed converter. The voltage controller takes the difference between the desired voltage and the actual output voltage as its input. It then generates a reference current for the load based on this difference. This reference current is limited intentionally to prevent the converter from using high current. The difference between the reference current and the actual current is then used as input for the current controller. The Proportional and Integral gain parameters of the Pi controller are denoted as Kp and Ki, respectively. The parameters Ki and Kp of the outer loop are set to be ten times faster than the parameters of the inner PI loop controller. The method used for tuning the controller parameters is a trial-and-error approach.

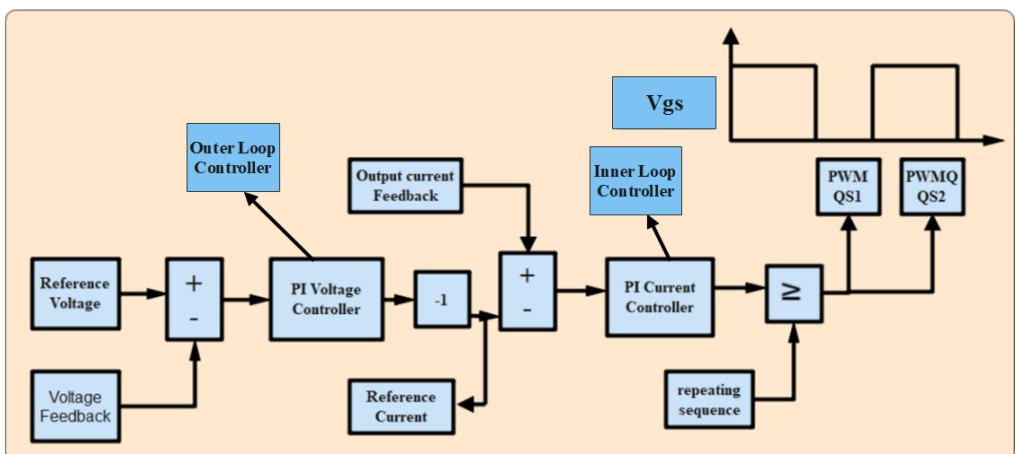

**Figure 7.** The suggested Controller strategy of the proposed Converter.

After applying the suggested dual PI controller to the proposed converter to verify the fixed output voltage under variable input voltage, Figure 8a shows that the output voltage of the proposed converter stays at 200 V even when the input voltage drops to its lowest values. This means that the suggested controller for the proposed converter is more reliable for applications that need a high consistent output voltage, and it can handle a wide range of duty ratios for higher power density in renewable energy applications. In Figure 8b, it is demonstrated that the converter can provide a variable output voltage ranging from 100 V to 250 V while keeping the input voltage fixed. It also responds quickly to changes, ensuring a swift adjustment in the output voltage.

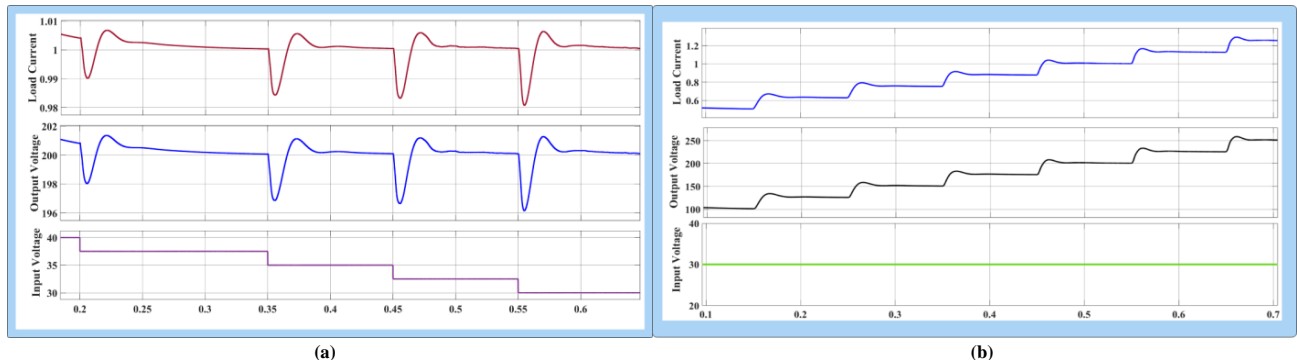

**Figure 8.** (**a**) output voltage of the proposed converter at variable input voltage, (**b**) variable output voltage at fixed input voltage.

## 8. Simulation and Experimental Results and Discussion

In this section, a 200 W PCB prototype is designed to validate the experimental results, as shown in Figure 9a. An experimental test is performed in the laboratory for the proposed converter, as shown in Figure 9b. Additionally, MATLAB Simulink and PLECS software are used to verify the experimental results in different cases.

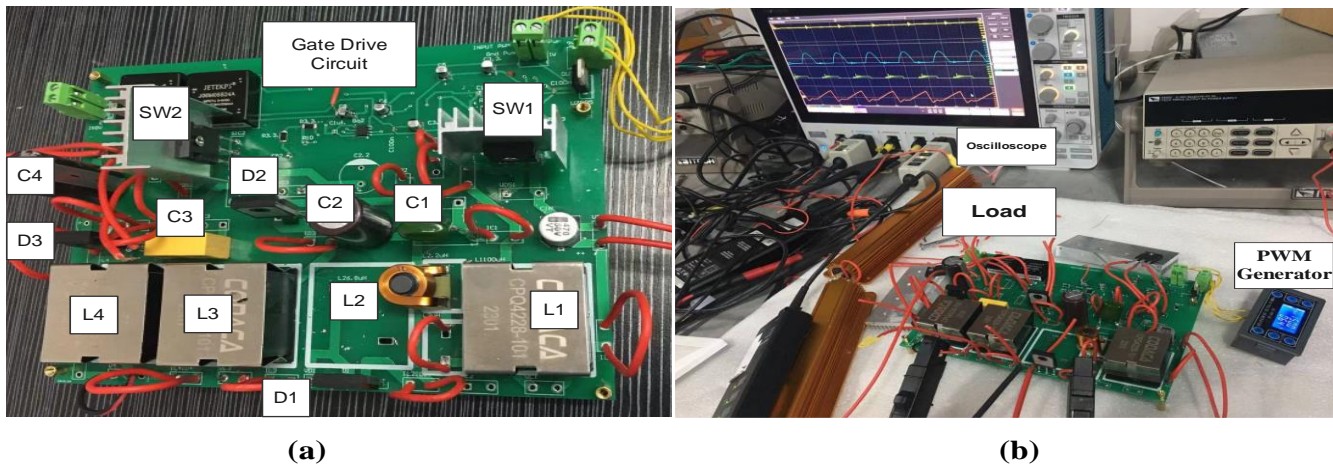

**(a)**

**(b)**

**Figure 9.** (**a**) PCB prototype of the proposed converter (**b**) experimental test of the proposed converter.

Figure 10a shows the source-to-gate voltage with a duty cycle of 27% and an output voltage of 218 V, with the input voltage at 40 V. It can be seen that the load current is 0.44 A at 95 W. Figure 10b shows the voltage across capacitors $C_1$, $C_2$, and $C_3$. It can be observed that $Vc_1$ is equal to the difference between $Vs$ and $Vc_2$, while the voltage across $C_2$ is 54.6 V. Furthermore, the average voltage across $C_3$ is zero. In Figure 10c, the current through switching $Sw_1$ and $Sw_2$ and the voltage across $Sw_1$ and $Sw_2$ are depicted. It can be seen that the current through $Sw_1$ is equal to $iL_1$ and $iL_2$, matching the shape of both inductor currents. Additionally, the voltage across the MOSFET significantly decreases when the converter operates at low duty cycle.

Figure 10d illustrates the current through $D_1$, $D_2$, and $D_3$. The current through $D_1$ is in series with $L_2$ to prevent $iL_2$ from starting in the reverse direction, having the same shape as the current in $L_2$. This prevents the input current from becoming pulsating at a low duty cycle. The current across $D_2$ is equal to $iL_2$ and $iL_1$ when $D_2$ is on, while the current through $D_3$ is equal to $iL_3$ and $iL_4$ when $D_3$ is on. Figure 10e displays the voltage across diodes $D_1$, $D_2$, and $D_3$. It can be observed that the voltage across power diodes significantly decreases when the converter operates at a low duty cycle. Figure 10f shows the voltage across inductors. The voltage across inductor $L_1$ is substantially reduced in the on state, equal to the input voltage, while in the off state, it is equal to half the input voltage. The voltage across $L_2$ in the on state has the same shape as $Vc_1$, and in the off state, $VL_2$ is equal to $Vc_2$ for a short period of time. $VL_3$ and $VL_4$ are equal to the value of $Vc_2$, which is very small during the on state and equal to the output voltage during the off state for a very short time as well. This means low voltage stress on the inductors, reducing the total losses of the proposed converter.

Figure 11a indicates that current flows through the inductors during the on-state period. It is evident that the discharging time of $L_2$ is smaller than that of $L_3$ and $L_4$. Additionally, the discharging time of $L_2$ is denoted as $\beta_1$ when the proposed converter operates in DCMC1. Furthermore, the discharging times of $L_3$ and $L_4$ are equal to D1, and both discharging times (D1 and $\beta_1$) depend on the values of RL, Vo, D. The inductor $L_1$ exhibits a long discharge time, does not reach zero, and shows no pulsation during the off states. Figure 11b illustrates the current through the inductors when the proposed converter operates in CCM. Figure 11c presents the load current of the proposed converter, which is 1 A at an input voltage of Vs = 20 V and an output voltage of 212 V at 200 W. Figure 11d

depicts an input voltage of Vs = 40 V and an output voltage of 208 V at a load current of 1 A at 200 W. It can be observed that the current through $L_1$ is still in CCM, while the current through $L_3$ and $L_4$ is in DCM at 200 W. This approach aims to reduce voltage stress across $Sw_2$ and inductors $L_3$ and $L_4$, thereby enhancing the performance and efficiency of the proposed converter. Figure 11e shows the voltage across $D_1$, and it can be seen that in the on state, the voltage across $D_1$ is around 22 V during the period (D − $\beta_2$ < t < D). $D_1$ works in ZCS, and inductor $L_2$ is an open circuit during this time. Therefore, when the input voltage is decreased to the minimum value, the proposed converter can work with high performance with one power diode working in ZCS, reducing the number of passive components. In addition, the voltage stress across $D_2$ will be reduced when the converter works in DCMC2 when capacitor $C_1$'s charge becomes zero at (D − $\beta_2$).

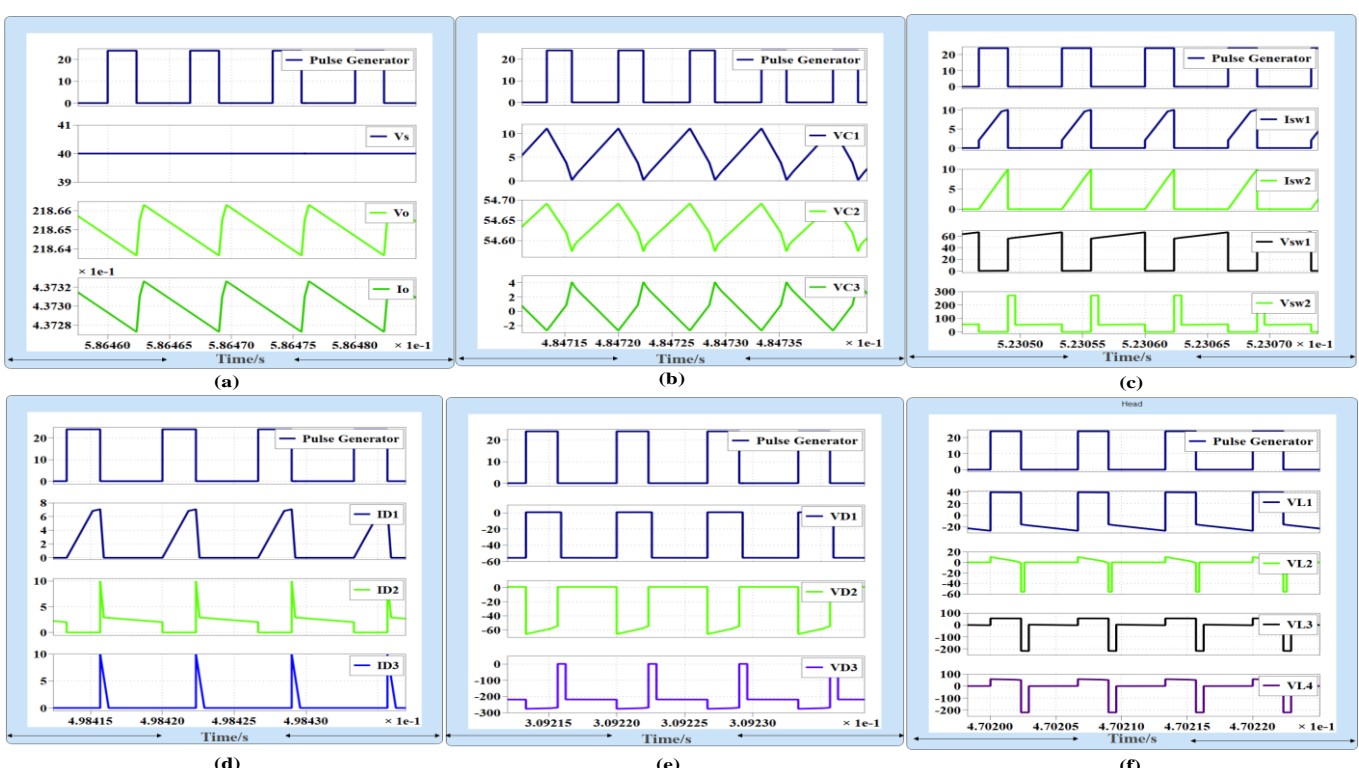

**Figure 10.** (**a**) Vs, Vo, Io, (**b**) $Vc_1$, $Vc_2$, $Vc_3$, (**c**) $Isw_1$, $Isw_2$, $Vsw_1$, $Vsw_2$, (**d**) $ID_1$, $ID_2$, $ID_3$, (**e**) $VD_1$, $VD_2$, $VD_3$, (**f**) $VL_1$, $VL_2$, $VL_3$, $VL_4$.

Figure 11f shows that the current reduction of $Sw_1$ occurs when capacitor $C_1$'s charge becomes zero at (D − $\beta_2$). After this period, the current through $Sw_1$ will come only from $L_1$. This means that the RMS current of $Sw_1$ will be significantly reduced when the converter operates at high current. The value of $\beta_2$ depends on the values of $C_1$ and $L_2$. This means that the proposed converter can supply high load current with high efficiency, especially for battery charging and renewable energy applications.

Figure 12a shows the inductor currents $L_1$ and $L_2$ when the converter operates in DCMC1 at D = 0.45. In Figure 12b, the inductor currents $L_1$ and $L_2$ can be observed when the proposed converter operates in DCMC2. Figure 12c demonstrates that the current through inductors $L_3$ and $L_4$ operates in DCM, with both inductors having the same value but in opposite directions. This method aims to reduce voltage stress across $Sw_2$ during the very long period of time ($D_1$ < t < Ts), as previously mentioned.

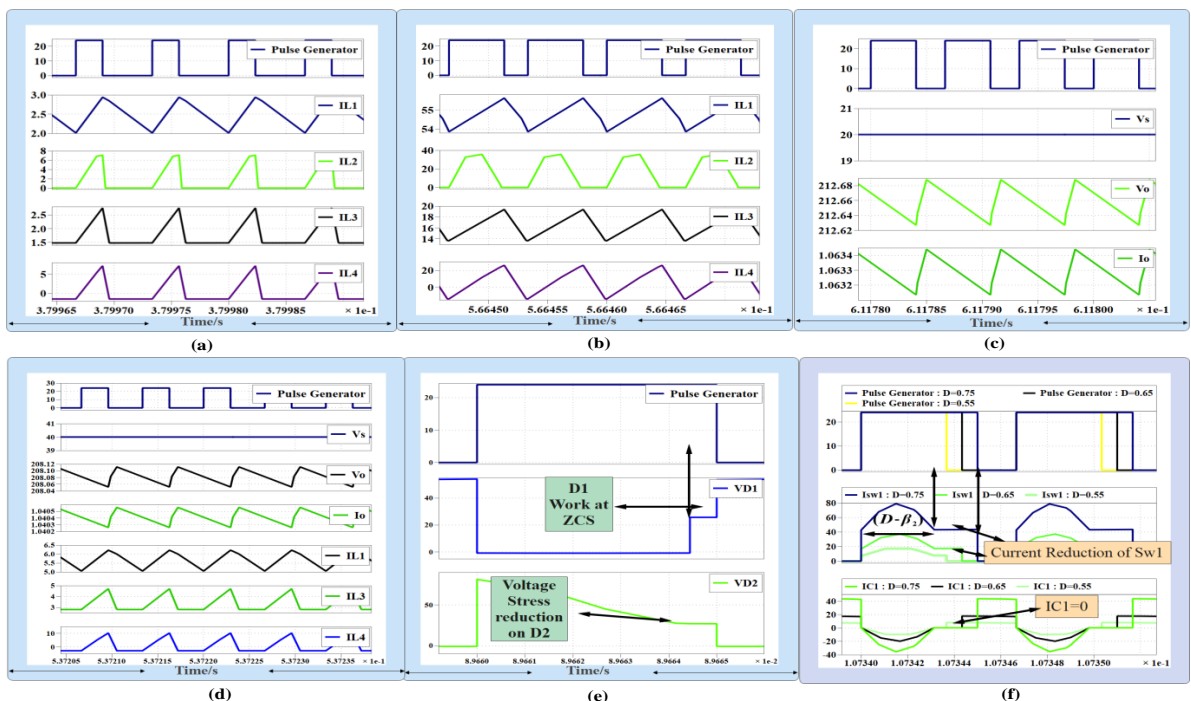

**Figure 11.** (**a**) iL$_1$, iL$_2$, iL$_3$, iL$_4$ at low duty cycle, (**b**) iL$_1$, iL$_2$, iL$_3$, iL$_4$ at CCM, (**c**) Vs, Vo, Io, (**d**) Vs, Vo, Io, iL$_1$, iL$_3$, iL$_4$, (**e**) VD$_1$, VD$_2$, (**f**) Isw$_1$ and IC$_1$ at different D.

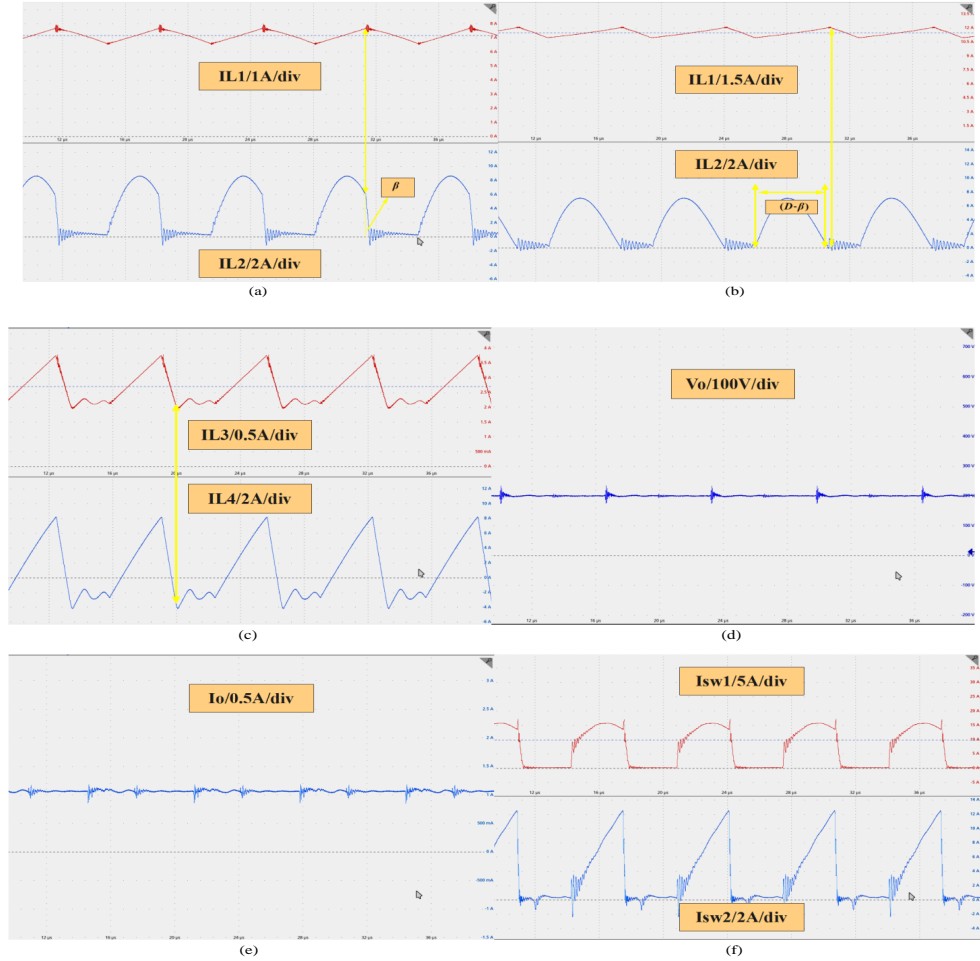

**Figure 12.** (**a**) iL$_1$, iL$_2$, (**b**) iL$_1$, iL$_2$, (**c**) iL$_3$, iL$_4$, (**d**) Vo = 200 V, (**e**) Load Current 1 A, (**f**) Isw$_1$, Isw$_2$.

In Figure 12d, the output voltage of the proposed converter is 200 V at a load current of 1 A, as shown in Figure 12e when the proposed converter operates in DCMC2. The currents through $Sw_1$ and $Sw_2$ are depicted in Figure 12f. It can be seen that the voltage across $Sw_1$ is equal to 54 V, which is a very low voltage as shown in Figure 13a. Figure 13b shows the voltage across $Sw_2$, which remains at 54 V for a long time period ($D_1 < t < T_s$). Figure 13c illustrates the voltage stress across $D_3$. Furthermore, Figure 13d shows the voltage across $D_1$, indicating that during the on state, the voltage across $D_1$ is around 22 V in the period ($D - \beta_2 < t < D$), with $D_1$ operating in ZCS, while the inductor $L_2$ acts as an open circuit during this time. As mentioned earlier, the voltage stress across $D_2$ will be reduced when the converter works in DCMC2, as capacitor $C_1$'s charge becomes zero at ($D - \beta_2$).

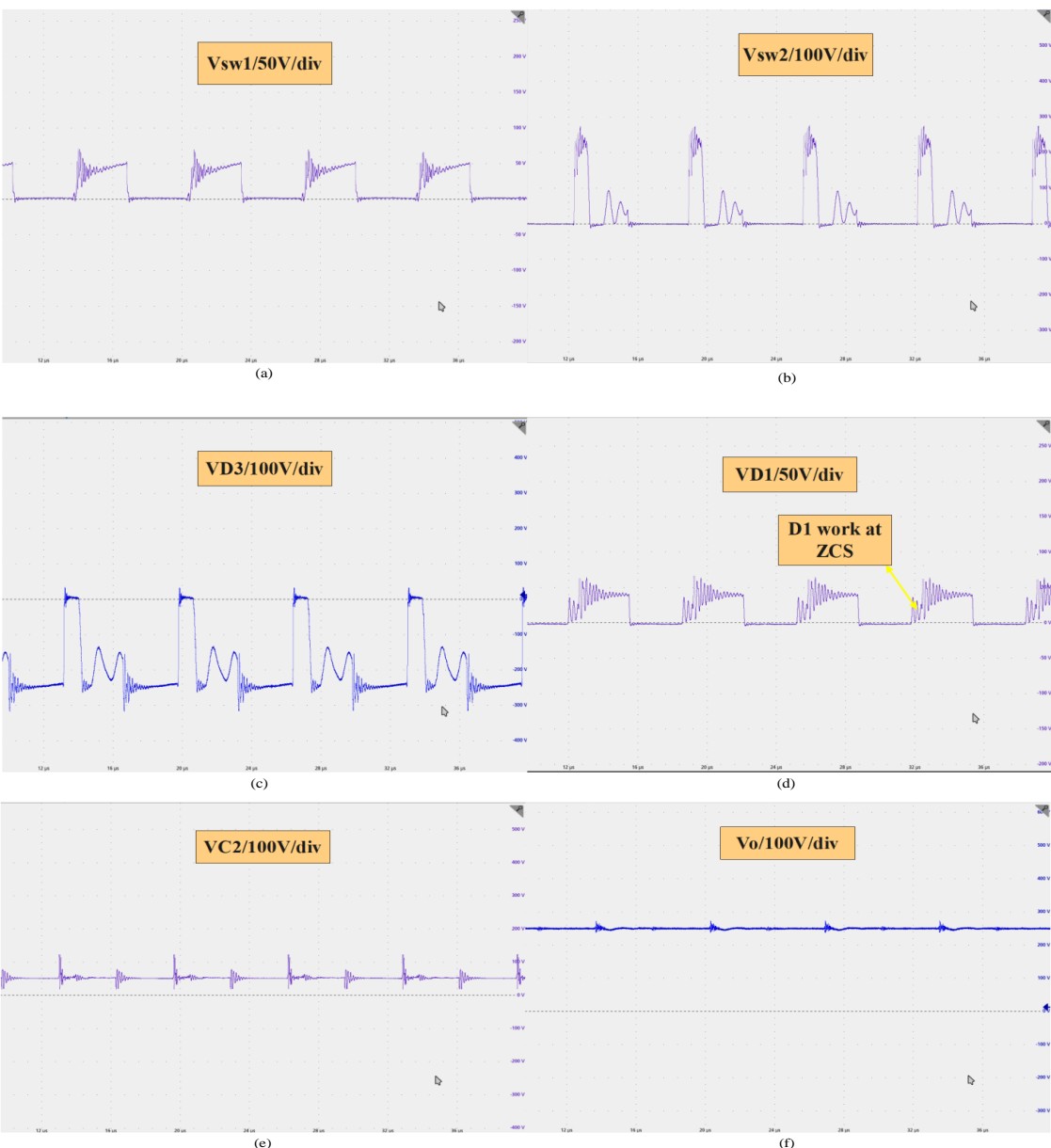

**Figure 13.** (**a**), $Vsw_1$, (**b**) $Vsw_2$, (**c**), $VD_3$, (**d**) $VD_1$ at ZCS, (**e**) $VC_2$, (**f**) Variable output voltage 250 V.

Figure 13e displays the voltage across $C_2$, which is equal to 52 V at D = 0.45. Additionally, the proposed converter can supply a variable output voltage, as shown in Figure 13f, where the output voltage is 250 V. These qualities make the suggested converter highly efficient and enable it to perform exceptionally well. Moreover, it can supply high currents, even when the duty ratios vary over a wide range.

The proposed converter demonstrates a higher level of performance compared to previous DC-DC converters. Notably, the power MOSFETs in the proposed converter experience significantly reduced voltage stress when operating in Discontinuous Conduction Mode (DCM) for both cases. Additionally, the single-cell switched inductor capacitor operates in resonant mode when the duty cycle exceeds 50%.

Furthermore, one of the passive elements, $L_2$, becomes an open circuit when the charge on capacitor $C_1$ reaches zero at $(D - \beta_2)$. This approach effectively reduces the voltage stress across the power MOSFETs and all diodes, while also minimizing the current stress on the main switch. This is achieved by ensuring that the current through the main switch is solely derived from $L_1$, which is equal to the input current.

The proposed converter also achieves a reduction in power losses, resulting in increased overall efficiency. Specifically, the efficiency of the converter reaches 96.5% at 200 W. Additionally, the input current does not reach zero at low duty cycles, making the proposed converter more efficient and well-suited for Renewable Energy Systems (RESs).

## 9. The Proposed Converter Efficiency Calculation

The proposed converter consists of four inductors, four capacitors, two power switches, and three diodes. These components, both passive and active, are not ideal. For example, an inductor has internal resistance, which increases as its value increases. The internal resistances of the inductors are denoted as $rl_1$, $rl_2$, $rl_3$, and $rl_4$. Similarly, the capacitors $C_1$, $C_2$, $C_3$, and $C_4$ have equivalent series resistances $rc_1$, $rc_2$, $rc_3$, and $rc_4$. The power diode also incurs power losses due to its internal resistance and forward voltage Vf. Additionally, power losses occur due to conduction and switching losses in the power MOSFET devices. Therefore, it is important to consider all of these losses for the proposed converter. The internal resistances of all active and passive elements are illustrated in Figure 14.

$$Irms = \sqrt{\frac{1}{Ts} \int_0^{Ts} (I)^2 dt} \tag{58}$$

$$Isw1rms = \sqrt{\frac{1}{Ts} \left[ \int_0^{(D-\beta_2)Ts} (iL1 + iL2)^2 dt + \int_{(D-\beta_2)Ts}^{D} (iL1)^2 dt \right]} \tag{59}$$

$$Isw2rms = \sqrt{\frac{1}{Ts} \int_0^{DTs} (iL3 + iL4)^2 dt} \tag{60}$$

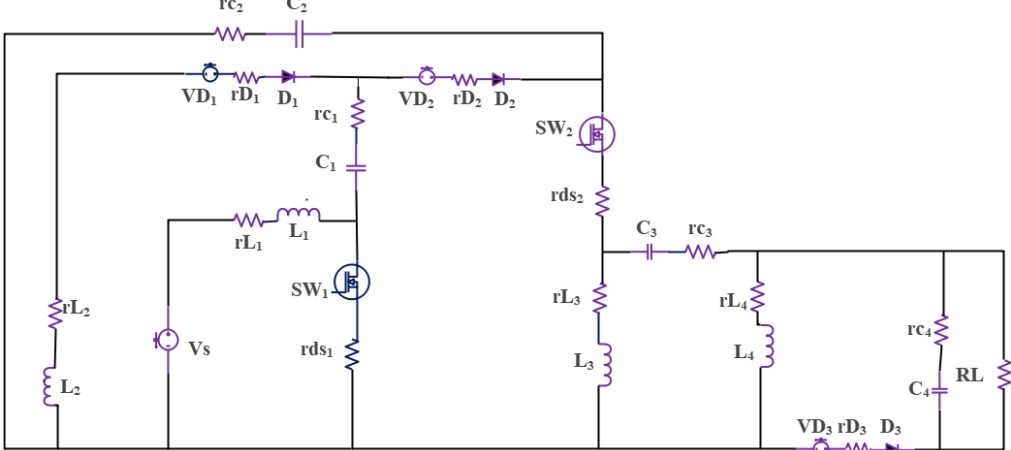

**Figure 14.** The proposed converter with parasitic components.

To calculate the total power losses of the proposed converter, the rms current is required for calculations related to the inductors, capacitors, power MOSFET, and diodes in both the on and off states. Equation (58) represents the general equation for rms current, and Equations (59) and (60) can be used to obtain the rms current through $Sw_1$ and $Sw_2$ during the on state.

$$ID1rms = \sqrt{\frac{1}{Ts}\int_0^{(D-\beta_2)Ts}(iL2)^2dt} = iL2rms \tag{61}$$

$$ID2rms = \sqrt{\frac{1}{Ts}\int_D^{Ts}(iL1)^2dt} \tag{62}$$

$$ID3rms = \sqrt{\frac{1}{Ts}\int_D^{Ts}(iL3+iL4)^2dt} \tag{63}$$

To calculate the rms current through power diodes, Equation (61) provides the rms current through $D_1$ during the on state, which is equivalent to the rms current through inductor $L_2$. Equations (62) and (63) describe the rms current across $D_2$ and $D_3$, respectively.

$$Ic1rms = \sqrt{\frac{1}{Ts}[\int_0^{(D-\beta_2)Ts}(iL2)^2dt + \int_D^{Ts}(iL1)^2dt]} \tag{64}$$

$$Ic2rms = \sqrt{\frac{1}{Ts}[\int_0^{DTs}(iL3+iL4)^2dt + \int_D^{Ts}(iL1)^2dt]} \tag{65}$$

$$Ic3rms = \sqrt{\frac{1}{Ts}[\int_0^{DTs}(iL4)^2dt + \int_D^{Ts}(iL3)^2dt]} \tag{66}$$

$$Ic4rms = \sqrt{\frac{1}{Ts}[\int_0^{DTs}(iL4)^2dt + \int_D^{Ts}(iL3+iL4)^2dt]} \tag{67}$$

To calculate the rms current through capacitors $C_1$, $C_2$, $C_3$, and $C_4$, the values can be determined from Equations (64), (65), (66), and (67), respectively.

$$Isw_1rms = \frac{IoD\sqrt{4D-3\beta_2}}{(1-D)^2} \tag{68}$$

$$Isw_2rms = \frac{Io\sqrt{D}}{(1-D)} \tag{69}$$

After solving the rms values in Equations (59) and (60), the resulting Equations (68) and (69) describe the rms current through $Sw_1$ and $Sw_2$, respectively.

$$ID_1rms = \frac{IoD\sqrt{D-\beta_2}}{(1-D)^2} = iL_2rms \tag{70}$$

$$ID_2rms = \frac{IoD}{\sqrt{(1-D)^3}} \tag{71}$$

$$ID_3rms = \frac{Io}{\sqrt{(1-D)}} \tag{72}$$

After solving for the rms values, Equations (70)–(72) provide the rms currents through the power diodes.

$$iL_1rms = \frac{IoD}{(1-D)^2} \tag{73}$$

$$iL_3rms = iL4rms = \frac{Io}{(1-D)} \tag{74}$$

Furthermore, Equations (73) and (74) provide the rms current through inductors $L_1$, $L_3$, and $L_4$. Equations (75)–(78) yield the rms current for capacitors $C_1$, $C_2$, $C_3$, and $C_4$, respectively.

$$Ic_1rms = \frac{IoD\sqrt{(1-\beta_2)}}{(1-D)^2} \tag{75}$$

$$Ic_2rms = \frac{IoD\sqrt{1+D(1-D)}}{\sqrt{(1-D)^3}} \tag{76}$$

$$Ic_3rms = Io\sqrt{\frac{D}{1-D}} \tag{77}$$

$$Ic_4rms = Io\sqrt{\frac{D}{1-D}} \tag{78}$$

### 9.1. Conduction Losses Calculation for MOSFET Devices

Conduction losses refer to the power losses that occur when current flows through a device, such as a power MOSFET or any other semiconductor device. These losses are primarily caused by the resistance of the device's conducting path, resulting in power dissipation in the form of heat.

To calculate the conduction losses of the power MOSFET in the proposed converter, the square value of the rms current is multiplied by the internal resistance of the MOSFET, as shown in Equation (79).

$$Pcd1 = \frac{PoD^2(4D-3\beta_2)}{RL(1-D)^4}rds1$$
$$Pcd2 = \frac{PoD}{RL(1-D)^2}rds2 \tag{79}$$

From Equation (79), the power conduction losses of power MOSFETs, Pcd1, and Pcd2, can be obtained.

### 9.2. Switching Losses Calculation for MOSFET Devices

Switching losses, also known as dynamic losses, are the power losses that occur during the switching transitions of a power electronic device, such as a power MOSFET. These losses result from the energy dissipated as the device switches between the on and off states. Switching losses are mainly caused by the charging and discharging of internal capacitances, as well as the voltage and current overlapping during the switching process.

To calculate the switching losses of the power MOSFET in the proposed converter, half of the square of the voltage stress across the MOSFET during the off state is multiplied by the switching frequency and the output capacitor of the power MOSFET (Co).

$$Esw = \tfrac{1}{2}CoVsw^2$$

$$P_{SW} = EswFs$$

$$P_{SWL1} = \frac{Vs^2D^2}{2(1-D)^2}FsCo$$

$$P_{SWL2} = \frac{Vs^2D^2}{2(1-D)^4}FsCo \tag{80}$$

where, (Esw) is the energy dissipated during one switching cycle. From Equation (80), the power switching losses of the MOSFETs $Sw_1$ and $Sw_2$, denoted as PSWL1,2, can be obtained.

*9.3. Total Power Loss in MOSFET Devices*

$$PTML1,2 = \frac{PoD^2(4D - 3\beta_2)}{RL(1-D)^4}rds1 + \frac{PoD}{RL(1-D)^2}rds2 + \frac{Vs^2D^2}{2(1-D)^2}FsCoFsCo + \frac{Vs^2D^2}{2(1-D)^4}FsCo \tag{81}$$

From Equation (81), PTML1,2 can be obtained as the total power losses of $Sw_1$ and $Sw_2$ by adding Equations (79) and (80).

*9.4. Losses in Power Diode*

Power losses in the diode can be divided into two components: losses due to internal resistance rd and losses due to the forward diode voltage Vf. Therefore, all power losses from the three diodes in the converter must be taken into account. From Equation (82), the average current (IDav) through diodes can be calculated:

$$\left.\begin{array}{l} I_{D1av} = \frac{PoD(D - \beta_2)}{RL(1-D)^2} \\[2mm] I_{D2av} = \frac{IoD}{(1-D)} \\[4mm] I_{D3av} = Io \end{array}\right\} \tag{82}$$

In order to find the power losses due to forward voltage (Pvf), Equation (82) is multiplied by Vf, resulting in Equation (83).

$$\left.\begin{array}{l} P_{Vf} = {}_{IDav}Vf \\[2mm] P_{Vf1} = Vf_1\frac{PoD(D - \beta_2)}{RL(1-D)^2} \\[2mm] P_{Vf2} = Vf_2\frac{IoD}{(1-D)} \\[4mm] P_{Vf3=}Vf3Io \end{array}\right\} \tag{83}$$

To calculate the diode power losses due to internal resistance (rd), the square of the diode's rms current is multiplied by rd, as shown in Equation (84).

$$\left.\begin{array}{l} P_{Dr} = IDrmsrd \\[2mm] P_{Dr1} = \frac{PoD^2(D - \beta_2)}{RL(1-D)^2}rd1 \\[2mm] P_{Dr2} = \frac{PoD^2}{RL(1-D)^3}rd2 \\[2mm] P_{Dr3} = \frac{Po}{RL(1-D)}rd3 \end{array}\right\} \tag{84}$$

The total power losses PDL1,2,3 across the three diodes can be found in Equation (85), by adding all the losses in the power diodes.

$$P_{DL1,2,3} = P_{Dr1,2,3} + P_{Vf1,2,3} \tag{85}$$

### 9.5. Power Losses in Inductors and Capacitors

The inductor used in the proposed converter has a very low internal resistance, as shown in Table 1. Additionally, the inductors and capacitors of the proposed converter are designed for high switching frequency, resulting in the converter operating with very high efficiency and performance. Calculation of the power losses due to the internal resistance of the inductors and capacitors are shown below:

$$
\left.
\begin{aligned}
PL &= iLrms^2 rl \\
PL1 &= \frac{PoD^2}{RL(1-D)^4} rl1 \\
PL2 &= \frac{PoD^2(D-\beta_2)}{RL(1-D)^2} rl2 \\
PL3 &= \frac{Po}{RL(1-D)^2} rl3 \\
PL4 &= \frac{Po}{RL(1-D)^2} rl4
\end{aligned}
\right\}
\tag{86}
$$

$$
\left.
\begin{aligned}
PC &= Icrms^2 rc \\
PC_1 &= \frac{PoD^2(1-\beta_2)}{RL(1-D)^4} rc_1 \\
PC_2 &= \frac{PoD^2(1+D(1-D))}{RL(1-D)^3} rc_2 \\
PC_3 &= \frac{PoD}{RL(1-D)} rc_3 \\
PC_4 &= \frac{PoD}{RL(1-D)} rc_4
\end{aligned}
\right\}
\tag{87}
$$

From Equations (86) and (87), the power losses $P_L$ and $P_C$ in the inductors and capacitors can be found, respectively.

### 9.6. Total Power Losses in Proposed Converter

The proposed converter losses can be divided into MOSFET losses, diode losses, inductor losses, and capacitor losses. The total power loss ($T_{PPCL}$) of the proposed converter can be found in Equation (88), which involves summing the power losses in the power MOSFETs ($P_{TML1,2}$), the total power losses in the diodes ($P_{DL1,2,3}$), and the losses in the inductors and capacitors ($P_{L1,2,3,4}$) and ($P_{C1,2,3,4}$), respectively. The proposed converter efficiency can be obtained using Equation (89).

$$
T_{PPCL} = P_{TML1,2} + P_{DL1,2,3} + P_{L1,2,3,4} + P_{C1,2,3,4}
\tag{88}
$$

$$
\eta = \frac{Po}{Po + T_{PPCL}} 100\%
\tag{89}
$$

The use of SiC MOSFETs with very low on-state resistance is a better choice to reduce conduction losses. Additionally, using inductors with low values and very low internal resistance can increase the performance and efficiency of the proposed converter.

Figure 15a shows that the proposed converter's output voltage increases as the switching frequency increases. This implies that the design of the proposed converter allows for boosting low voltage to high voltage at a low duty cycle, achieving high power density with an efficiency of 96.5%. Figure 15b provides a comprehensive visual representation of the losses incurred by the proposed converter. It is worth noting that a significant proportion of these losses can be attributed to the power MOSFETs and diodes used in the converter. These losses result from the switching and conduction characteristics exhibited by these components during the converter's operation. Furthermore, a portion of the losses can be attributed to the inherent resistance of the inductors and capacitors present in the system.

Figure 16a,b illustrates the conduction power losses of MOSFETs $Sw_1$ and $Sw_2$ at different input voltages. It can be observed that the conduction power losses for both MOSFET switches slightly increased as the input voltage rose from 10 V to 40 V. Additionally, the conduction losses showed a significant decrease as the duty cycle decreased.

This indicates that the SiC MOSFET utilized in the proposed converter exhibits very low power conduction losses, particularly at variable duty cycles. Consequently, the proposed converter achieves higher efficiency compared to previous DC-DC converters. By employing WBG (Wide Bandgap) MOSFETs, both conduction and switching losses can be substantially reduced, leading to a significant increase in the converter's efficiency.

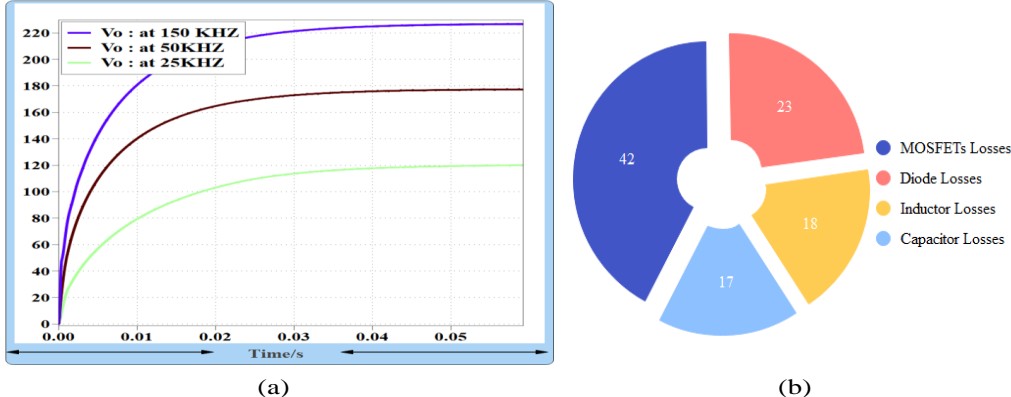

**Figure 15.** (**a**) Output voltage of the proposed converter at different switching frequency at D = 45% (**b**) percentage losses of components of proposed converter.

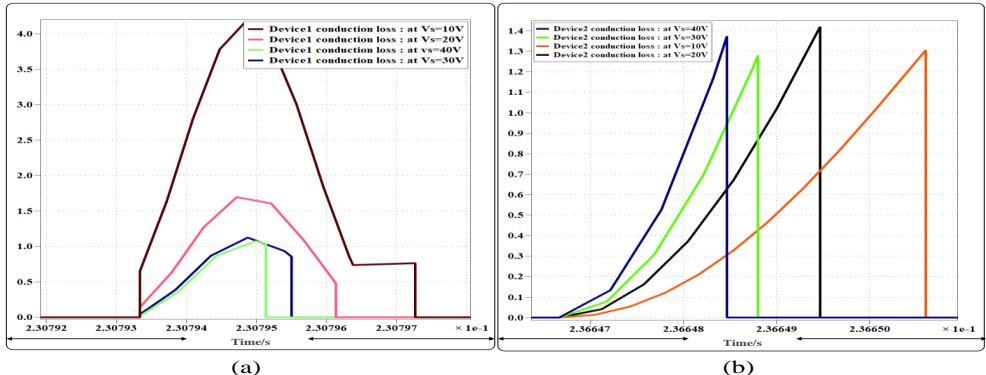

**Figure 16.** Conduction losses of power switches at variable input voltage proposed converter (**a**) power Mosfet 1 (Sw$_1$) (**b**) power Mosfet 2 (Sw$_2$).

Figure 17a,b displays the efficiency of the proposed converter at various input voltages. The results indicate that at an input voltage of 30 V and a duty cycle of 50%, the converter achieves an efficiency of approximately 97.3% when operating at a power level of 200 W. This demonstrates that the proposed converter is capable of stepping up a low input voltage under light load current conditions, while also providing high load current at the maximum input voltage.

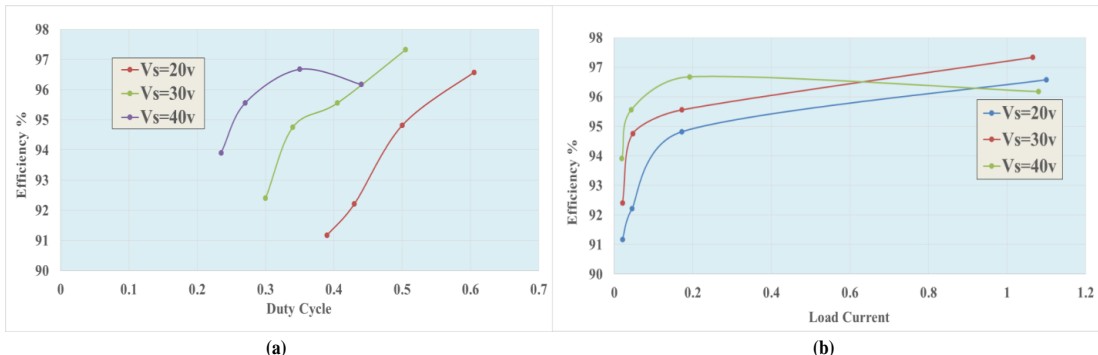

**Figure 17.** (**a**) Efficiency of the proposed converter with duty ratio at variable input voltage (**b**) Efficiency of the proposed converter with load current at variable input voltage.

## 10. Conclusions

As a result, a new single-cell hybrid switched inductor DC-DC converter is proposed to demonstrate the verification of ultra-high voltage gain in photovoltaic applications. The modification in the proposed converter helps prevent the input current from pulsating to zero at very low duty cycles, making it more efficient for renewable energy applications. The single cell of the hybrid inductor is interleaved with the main switch to reduce current stress when the load current increases and the capacitor charge becomes zero. Moreover, the addition of a modified hybrid switch inductor with a capacitor, operating in dual boosting mode with a single switch, allows the converter to achieve ultra-high voltage gain.

The proposed converter offers several advantages, including ultra-high voltage gain, high efficiency, low voltage stress on power MOSFETs, diodes, inductors, and capacitors, as well as low switching and conduction losses. Furthermore, the proposed converter utilizes transformerless and non-coupled inductors. Additionally, the proposed converter's efficiency is around 96.5% when the input voltage is 20 V with a duty cycle of 0.6. The increased flexibility in the duty cycle allows the proposed converter to operate at high power density and convert very low input voltage to high output voltage for renewable energy systems.

In addition, the output voltage of the proposed converter increases when the switching frequency is increased to boost low input to high output voltage at a low duty cycle. The voltage stress on the power devices has been reduced compared to existing DC-DC converters. Moreover, using a high switching frequency reduces the component values and circuit size, resulting in a significant reduction in the weight of the proposed converter. Passive components of the proposed converter are reduced when the converter operates in (DCM) and (CCM), which improves converter efficiency and performance.

**Author Contributions:** Conceptualization, X.W.; Software, A.F.A.; Formal analysis, A.F.A.; Writing—original draft, A.F.A.; Writing—review & editing, A.F.A.; Supervision, X.W. All authors have read and agreed to the published version of the manuscript.

**Funding:** This research received no external funding.

**Data Availability Statement:** The data presented in this study are available on request from the corresponding author.

**Conflicts of Interest:** The authors declare no conflict of interest.

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
