# Peer review of "A New Single-Cell Hybrid Inductor-Capacitor DC-DC Converter for Ultra-High Voltage Gain in Renewable Energy Applications"

_electronics, doi:10.3390/electronics12143101_

Round 1

Reviewer 1 Report

Abstract

Elucidate the concept of the "novel single cell hybrid inductor DC-DC converter."

Clarify why "ultra-high voltage gain" is critical in renewable energy applications.

Detail how the diode and double capacitor contribute to achieving voltage gain.

Define "current stress."

Provide more context about the operation of the "modified hybrid switch inductor with a capacitor."

Expand on specific benefits or results for each advantage of the proposed converter.

Include additional information about the experimental results.

Introduction

Condense complex points and move less critical information to subsequent sections.

Make the structure more coherent - provide a broad context, discuss specific problems, and introduce the proposed solution.

Group citations into categories rather than citing individually.

Rectify minor grammatical errors and awkward phrasing.

Clearly outline the proposed work's goals, approach, and potential benefits.

Include brief explanations of key terms for readers unfamiliar with the topic.

Enhance the explanation of the study's motivation and its potential impact.

Section 2

Improve clarity by simplifying language and providing further explanations of complex concepts.

Enhance organization by introducing clear headers and avoiding abrupt transitions.

Be consistent with terminology.

Refer to figures and tables in the text before they appear.

Ensure all mathematical symbols are clearly defined.

Summarize the main points at the end of each section.

Cite sources when referencing methods or technologies used.

Sections 3-7

Include explicit forms of all equations and discuss them contextually.

Provide all referred figures, ensuring they are well-labeled and discussed.

Maintain consistency in notation.

Compare with previous converters, discussing why your converter outperforms them.

Proofread the document for grammatical errors, complex sentences, and jargon.

Provide a brief introduction to key concepts.

Discuss potential real-world applications and acknowledge any limitations.

Section 8

Directly reference figures in the discussion.

Provide additional clarification or definitions for certain terms.

Explain the mathematical models or equations used in Table 2.

Improve sentence structure for better clarity.

Avoid repetition of information across figures.

Provide a clear conclusion summarizing the major findings.

Discuss the referenced works in Table 2 and how your work differs.

Proofread for minor grammatical and punctuation errors.

Sections 9-10

Simplify complex mathematical representations or explain their significance.

Ensure diagrams are clear, labeled, and adequately explained.

Define terms such as 'power MOSFET,' 'conduction losses,' 'switching losses', and variables used in equations.

Compare your results to other existing solutions in the field.

Discuss potential sources of error in your calculations.

Provide a clear conclusion and discuss potential future research.

Maintain consistency in tenses.

Cite all external sources, equations, methods, or theories.

Simplify complex sentences for better readability.

The English quality of this paper is generally satisfactory, but it could benefit from careful editing to address minor grammatical errors and enhance clarity.

Reviewer 2 Report

Dear Authors,

please find my comments on your paper listed below:

1) in the introduction, it is stated "The main problems are a lower count of inductors and capacitors..." Please add information about what number are you comparing with. Secondly, justify, why it is a problem. A lower number of components helps miniaturization and decreases costs and also a lower number of components introducing losses in power conversion, which may lead to higher efficiency.

2) Please divided the text into paragraphs. Manuscript in its current form is hard to read.

3) In the introduction, "A low switching frequency is used with high values of passive elements, significantly reducing efficiency.". Actually, low switching frequency favours higher efficiency, because switching losses are proportional to switching frequency. Please comment on this issue.

4) In section 4, equations allowing the calculation of voltage stresses on diodes and transistors are presented. However, in the real application, these stresses are influenced by parasitic inductances resulting in overvoltages during turning-off, which according to the paper [1], also influences the crucial operation parameter of any semiconductor device - its junction temperature. Authors should also include information about current stresses, including overcurrents.

[1] 10.1109/TPEL.2022.3180170 

5) Section 8, "Additionally, MATLAB Simulink and PLECS software are used to verify the experimental results in different cases." Please extend comments on simulations, used models and their accuracy. As results from the paper [2], using different models and simulation software in the simulations of the same converter may provide different values of its efficiency.

[2] 10.3390/electronics10232920

6) Equation 80 confused me. Typically, e.g. as in [1] or in PLECS [3], switching losses are calculated using Eon and Eoff parameters. Please justify why you did it in a different way.

[3] 10.1109/TIE.2022.3189102 

7) Fig. 7 - please state for which conditions and how you obtain results presented in this figure.

8) In the multiple parts of the manuscript, two different values of efficiency of the proposed converter are presented - 97.3% and 96.5%. Please unify them.

Inappropriate use of capital letters in the middle of a sentence and passive voice in multiple places, Incorrect selection of words like e.g. electronic elements instead of electronic components.

Reviewer 3 Report

This paper proposes a novel single cell hybrid inductor DC-DC converter to demonstrate the verification of ultra-high voltage gain in renewable energy applications. However, some descriptions are not clear. Some revisions are necessary in the manuscript.

1. Please indicate whether the article has further improved the control strategy.

2. Please make sure all parameters are defined.

3. Please add whether the expression in the article has the corresponding basis.

4. Please further highlight the core innovation of the article.

5. In the paper, authors have focused on DC–DC converters to step up the voltage for numerous applications such as street lights, motor drives, microgrid applications, uninterruptible power supplies, fuel cells, and medical equipment. Comparison of different applications such as microgrid applications needs to be analyzed to indicate advantages of your work, which can refer to:

[a] IEEE Transactions on Power Systems, vol. 37, no. 5, pp. 4067-4077, 2022

[b] IEEE Transactions on Computer-Aided Design of Integrated Circuits and Systems, vol. 41, no. 3, pp. 516-529, 2022

[c] IEEE Transactions on Industry Applications, vol. 58, no. 6, pp. 7952-7965, 2022

A proof reading is needed.

Round 2

Reviewer 1 Report

Thank you for your time and effort in responding to the review comments. However, upon reviewing your responses, I found them to be rather general and vague, not addressing the issues raised in a detailed, point-by-point manner as is customary in the review process.

For the review to proceed effectively, it is imperative to respond specifically to each point raised, as this ensures clarity on how the issues have been addressed. The lack of specific responses to each end of the review has made it difficult to assess where and how revisions have been made to the manuscript.

Therefore, I kindly ask that you revise your responses to provide a clear, point-by-point rebuttal. Please indicate the exact location of each revision in the manuscript. This would significantly aid in the evaluation process, helping to delineate exactly what changes have been implemented in response to the comments.

Your cooperation in this matter is much appreciated. I look forward to receiving your revised responses.

The English sentences are quite satisfactory.

Reviewer 2 Report

Dear Authors,

Thank you for your clear responses to all of my comments.

Round 3

Reviewer 1 Report

The responses provided by the authors seem to be appropriate and address the reviewer's comments and suggestions. The authors acknowledge the feedback, make specific revisions, and explain the changes they have implemented in response to each point raised by the reviewer. They have also provided additional explanations and clarifications and made necessary formatting adjustments to improve the readability and clarity of the manuscript. The authors have taken the reviewer's comments seriously and tried to address them satisfactorily.